# Designing Sustainable Drainage Systems as a Tool to Deal with Heavy Rainfall—Case Study of Urmia City, Iran

Reza Mehdizadeh Anvigh [1,*], José Figueiredo Silva [2] and Joaquim Macedo [1]

1   RISCO—Research Center for Risks and Sustainability in Construction, Department of Civil Engineering, University of Aveiro, 3810-193 Aveiro, Portugal; jmacedo@ua.pt
2   Department of Environment and Planning, University of Aveiro, 3810-193 Aveiro, Portugal; jfs@ua.pt
*   Correspondence: mehdizadeh@ua.pt

**Abstract:** Heavy rainfall, a natural phenomenon reinforced by climate change and global warming, can cause severe social, economic, and safety impacts. Due to the impact of climate change and global warming, heavy rainfall events have become more frequent and intense in recent years, underscoring the urgent need to develop robust stormwater management systems that can prevent related social, economic, and safety issues. This is of greater importance in developing countries. The present study identified areas in Urmia City, Iran, that require stormwater management to develop a comprehensive understanding of the hydrological processes within the study area and to prevent the subsequent effects of heavy rainfall. For this purpose, a combination of the watershed modeling system (WMS) and stormwater management model (SWMM) was employed. Also, three possible scenarios that could be implemented to address the issue of water flow in the medium were proposed. Results indicated that the scenario involving the application of a vegetative swale was the most promising solution. Overall, the results of the present study offer a valuable framework for decision-makers in regions facing heavy rainfalls to effectively manage and minimize the adverse impacts of such events.

**Keywords:** drainage systems; heavy rainfalls; sustainability; Urmia City; watershed modeling system (WMS) and stormwater management model (SWMM); simple additive weighting method (SAW)

## 1. Introduction

Flooding during heavy rainfall is a natural phenomenon that occurs when the amount of precipitation surpasses the capacity of the soil to absorb it [1]. This commonly occurs in many areas of the world and has significant impacts on the environment and human activities [2]. According to statistics, heavy rainfall events have become more frequent in recent years, and the trend is expected to continue due to climate change. The US National Oceanic and Atmospheric Administration (NOAA) has reported that the number of heavy precipitation events in the US has increased in recent decades. According to NOAA, the amount of precipitation falling in the heaviest 1% of rain events has increased by 71% in the continental US since 1958 [3]. The European Environment Agency (EEA) has also reported an increase in the frequency and intensity of heavy precipitation events in Europe. According to the EEA, the number of heavy precipitation events has increased by about 20% in Europe since the 1950s [4]. There are several reasons for the heavy rainfalls. However, the main causes include changes in atmospheric circulation patterns, increased water vapor content in the air, and changes in land use. Deforestation, urbanization, and other human activities that alter the natural landscape have also been found to contribute to heavy rainfall impact by changing the way water is absorbed and distributed in the environment [5–7].

Heavy rainfall can pose safety hazards to individuals and communities, such as flooding, landslides, and mudslides resulting in damage to property and infrastructure. It can also have severe environmental impacts, such as soil erosion, water pollution, and

damage to ecosystems. To mitigate these risks, it is essential to have effective emergency response plans, proper infrastructure for stormwater management, and proactive measures to reduce the impact of heavy rainfall on the environment [8]. To mitigate the impact of heavy rainfalls, countries have adopted various strategies based on different factors, such as their geographic location, climate conditions, and available resources [9,10]. In some countries, structural measures such as building dams, levees, and drainage systems have been implemented to control heavy rainfall and prevent flooding [11]. These measures have proven effective in some cases, such as in the Netherlands, where the use of dams and levees has prevented catastrophic flooding despite the country's low-lying geography. However, these measures can be costly and may not always be practical, particularly in developing countries. Additionally, some countries have adopted nature-based solutions, such as reforestation [12], wetland restoration, and the use of green infrastructure, to mitigate the impact of heavy rainfall [13]. These approaches have been shown to be effective in reducing the risk of flooding and erosion while also providing other environmental benefits, such as habitat restoration and carbon sequestration.

In response, many researchers and practitioners have proposed sustainable drainage systems as a potential solution to mitigate these effects. The implementation of sustainable drainage systems is advantageous over other methods as it provides rapid and efficient removal of stormwater runoff, which can help prevent flooding and minimize damage to urban infrastructure [14]. In addition, drainage systems provide other benefits such as improving water quality by removing pollutants from stormwater runoff and providing a reliable source of water for irrigation and other uses [15]. Several studies have shown the effectiveness of drainage systems in mitigating the negative effects of heavy rainfall events such as reducing the damage caused by flooding during a heavy rainfall event [16,17]. However, drainage systems also have some limitations and challenges. They can be expensive to design, construct, and maintain, and may require large amounts of land for construction. In addition, conventional drainage systems can sometimes exacerbate downstream flooding and erosion. Overall, drainage systems offer a critical solution for mitigating the negative effects of heavy rainfall events and can provide a range of benefits beyond flood mitigation. However, their design and implementation must take into account the specific characteristics of the urban area and the potential environmental impacts. In this regard, proper design of the vulnerable points and implementation of sustainable design concepts can boost the capacity of this measure in preventing the adverse effects of heavy rainfalls.

The current research implemented a blend of the watershed modeling system (WMS) and stormwater management model (SWMM) for hydrological analysis and hydraulic simulations, respectively, to pinpoint regions in Urmia City, Iran, that necessitate stormwater management improvement. This was completed to obtain a thorough comprehension of the hydrological processes within the study area and assess the consequent impacts of intense rainfall. After identifying the vulnerable areas (using WMS), three potential solutions were proposed to address the water flow issue in the medium. The study established a decision matrix using a simple additive weighting method (SAW) to prioritize influential criteria and objectively determine the most effective scenario.

Urban drainage systems have long been used as effective solutions to meet socio-hydrological demands in regard to protecting against flooding and complying with environmental concerns [18]. In this regard, Sustainable Urban Drainage Systems (SuDSs) have been adopted to reduce the negative impacts of urban development and related problems on local hydrology [19]. Sustainable drainage systems have been extensively used for runoff control whose application has gone beyond functional aspects to include other social aspects such as responding to the needs of the local community, protecting the quality of water, recharging the groundwater, and preserving the wildlife habitat while contributing significantly to urban aesthetics and enhancement of amenities [20–22].

Because of the challenges in designing and implementing sustainable drainage system models on rainfall control, there are a few studies in this regard. Therefore, no ample

evidence is found of its effectiveness. The effectiveness of green infrastructure for enhancing livability in different cities was explored through adapting effective measures such as mitigation of stormwater issues and promotion of water quality. The study used a decision-making framework in order to evaluate its application for Storm Water Management Model (SWMM) runoff simulations across Detroit, Michigan, and Addis Ababa, Ethiopia, and indicated that green infrastructure systems were effective for stormwater management in urban areas [23]. The relationship between urban drainage systems design and assessment of the effects of climate change on rainfall at local levels was studied in Ontario, Canada. Using extreme rainfall (ER) data, the study demonstrated that effective drainage systems had an acceptable performance in managing and mitigating the effects of stormwater [24]. Management of surface runoff in the northeast of England through sustainable drainage systems (SuDSs) was examined and provided evidence for its effective performance in controlling extreme rainfalls and treating surface water [25].

Studies in the literature have used different scenarios and approaches to control peak flows and mitigate the volume of the stormwater. Traditionally, stormwater detention tanks (SWDTs) have been used at critical catchments, but they are not cost-effective and cannot be implemented everywhere [26]. Sustainable drainage systems have been found to have a significant hydrological performance when integrated with other approaches like SWDTs and GI (green infrastructure) and have shown to be effective in reducing overflow scenarios [27]. Similarly, a resilience-based analysis of urban drainage systems (UDSs) at functional and structural failure modes was conducted in India. The study used the SWMM and different rainfall and urban growth scenarios and revealed a positive correlation between urban development and UDS and revealed that the drainage system was functionally and structurally resilient under simulated rainfall scenarios [28]. However, there are some challenges in this regard. It was found that SuDSs should be implemented with more care in developing countries for stormwater runoff because in such settings the potential for pathogenic factors is higher [29].

Despite all the challenges in evaluating and monitoring SuDSs, recent research has seen an increase in examining its effects on runoff. Monitoring SuDS performance often tends to be qualitative and is hindered by budgetary constraints. McDonald's study in Scotland revealed that SuDS monitoring and evaluation practices are largely informal and sporadic [30]. Furthermore, when monitoring and evaluation do occur, they typically involve descriptive methods, ranging from occasional site visits for photography to basic routine maintenance checks [31]. Moreover, there is a limited number of studies that quantitatively measure SuDS performance, particularly in urban settings or retrofit scenarios, largely due to the complexities involved in designing and executing such investigations [32,33]. The absence of systematic monitoring and evaluation leaves uncertainty regarding whether SuDS systems are operating optimally or experiencing deficiencies.

A study by [34] examined the use of hydraulic models to predict flood risk in urban areas. The study found that hydraulic models can be effective in predicting flood risk and identifying areas where mitigation efforts are most needed. Similarly, a study by [35] explored the use of hydraulic modeling to assess the potential for flood damage in a river basin. The study found that hydraulic modeling can provide useful insights into the potential for flood damage and help identify areas where risk reduction measures can be most effective.

The study by [36] explored the influence of various urban land use factors on waterlogging in Shenzhen City, focusing on both horizontal and vertical patterns at multiple scales (1–5 km). Employing Pearson correlation analysis and the random forest model, the research investigated key factors such as building coverage ratio, building crowding degree, building density, the proportion of impervious surfaces, the proportion of green space, and population density. The findings revealed that these factors significantly impact waterlogging density. Notably, horizontal patterns had the greatest influence at a 3 km scale, while the impact of vertical patterns, like building congestion, increased with scale.

The study implied that effective waterlogging mitigation requires controlling building density and planning urban patterns thoughtfully.

The study by [37] conducted a systematic review of existing methodologies for designing road network-based flood passage systems. It discussed new technologies to enhance system resilience and identified current knowledge gaps and future research directions. Key findings suggested that flood management measures should integrate accessibility assessment, lifeline, and emergency planning to ensure human well-being. Special attention was also highlighted to be given to the needs of vulnerable groups during the planning stage. Moreover, a data-driven approach was recommended for real-time management and evaluation of future works.

The present study explores the feasibility and effectiveness of SuDSs as a solution to manage excess rainfall in a developing country, the case of Urmia City in Iran, focusing on factors such as local topography, infrastructure, and socio-economic conditions. It offers some effective scenarios to manage the issue based on the factors that were identified. The novelty of the study lies in its innovative approach to adopting a low-cost and effective measure for mitigating the negative impacts of the issue while contributing to environmental concerns in developing countries that struggle with sustainability issues.

## 2. Materials and Methods

### 2.1. Study Area

Defining the study area is a crucial step in hydrology because it establishes the spatial boundaries within which hydrological processes are analyzed and modeled. The study area typically refers to the watershed or catchment, which is the land area that directs runoff water to a specific river or stream channel. For the present study, an area of 870 hectares of Urmia City in the Iranian province of West Azerbaijan was selected. In the study area, the dominant slope is from south to north. Hence, maximum flow transfer is expected in this direction. A primary site visit revealed some issues related to the development of urban drainage systems and stormwater management, listed below:

(a)  Lack of organization in non-urban sectors;
(b)  Lack of efficient drainage systems in most of the streets in the study area (Figure 1);
(c)  Lack of sustainable drainage systems designed in order to enhance the flow capacity in the study area (Figure 2);
(d)  Sudden change in the dimensions (tightness) (Figure 3).

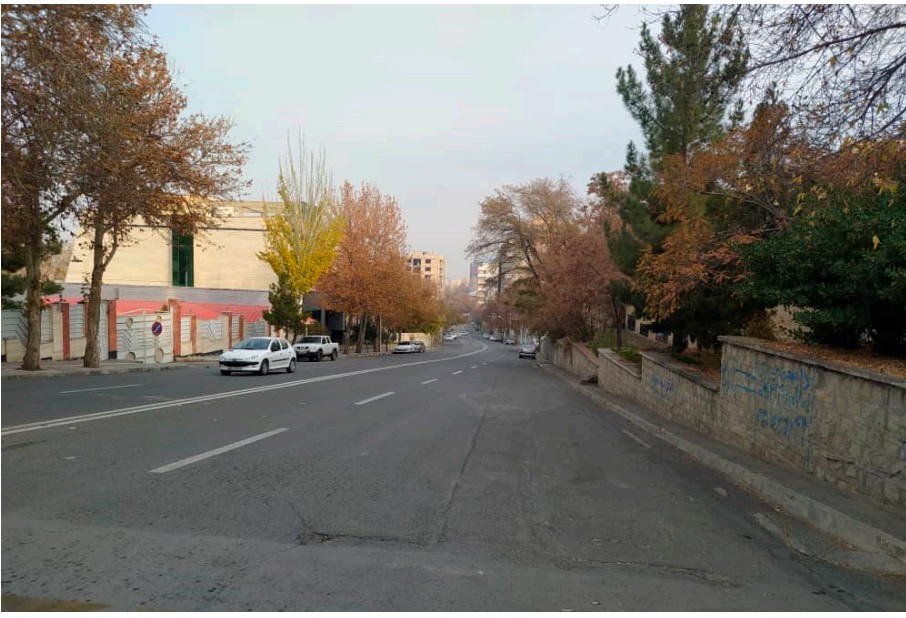

**Figure 1.** Lack of efficient drainage systems.

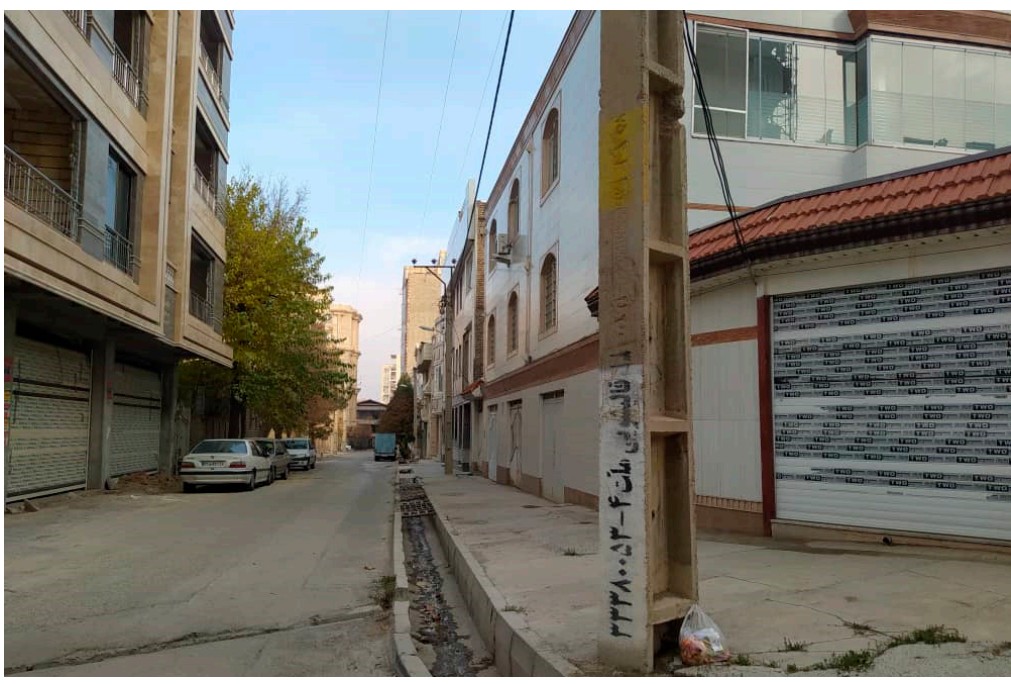

**Figure 2.** The lack of two-way drainage systems in most of the streets in the study area that can result in the inefficient collection of the rainfalls.

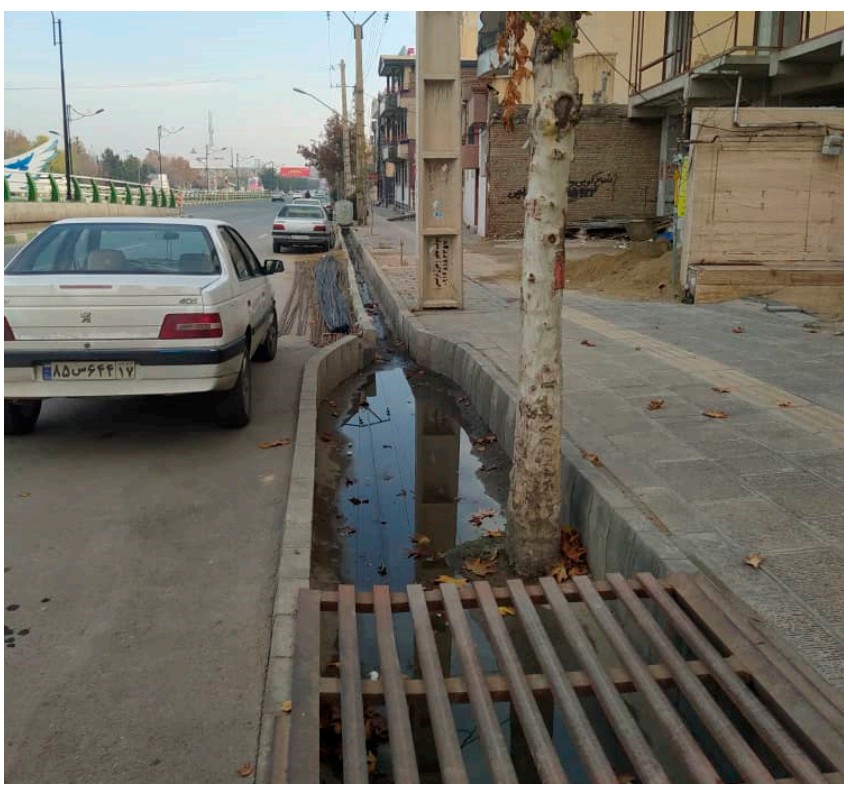

**Figure 3.** Sudden change in the dimensions (tightness).

## 2.2. Watershed Modeling System (WMS)

WMS 7.1, a simulation and modeling software that can be integrated with a geographic information system (GIS), is used to create digital watershed models that can simulate and analyze surface water hydrology, including rainfall–runoff processes, flood forecasting, water quality, sediment transport, and groundwater interactions [38]. The software allows

users to import, manipulate, and analyze digital elevation models, aerial photographs, and other spatial data to develop hydrological models of a given watershed. To proceed with the WMS model, the digital elevation model (DEM) maps, made using the topographic data of the study area, were imported into the model, and calculations were performed to draw the waterway network model. Then, a shapefile (SHP) layer was used to define the area and the A, B, and C points to calculate the specific physiographic model of the basin. Finally, the flood hydrograph model at those points was calculated.

WMS software was also used to calculate the flood hydrograph, using the SCS method 7.1. For this, the first step was to calculate the time interval and the respective calculation method including the loss method, and the lag time. Considering that there is no hydrometric station in the study area, the rational method of adjusting the runoff coefficient values in the calibration procedure was used for validation of the results of peak flow achieved in this study.

The Kerby–Kirpich method was used to estimate the watershed time of concentration according to [39]. The total time of concentration was calculated by the sum of the overland flow time (Kerby) and the channel flow time (Kirpich):

$$t_c = t_{ov} + t_{ch} \tag{1}$$

In this equation, $t_{ov}$, and $t_{ch}$ are the overland flow time (h), and the channel flow time (h), respectively. The $t_c$ calculated by this method applies to those watersheds ranging from 0.65 to 388 km$^2$, main channel lengths between 1.6 km and 80.5 km, and main channel slopes between 0.002 and 0.02 (cm/cm). According to the rainfall duration frequency (RDF) diagram for Urmia City, Iran, the soil conservation service (SCS) artificial rainfall curve was drawn using the stormwater management and design aid software (SMADA) software 6.43 using the SCS method.

### 2.3. Modeling with Stormwater Management Model (SWMM)

In the present study, the stormwater management model (SWMM) was implemented to simulate and model the process of urban flooding. SWMM 5.2.4. is a popular model that has been widely used as a dynamic rainfall–runoff simulation tool to calculate the water quantity and quality resulting from rainfall conditions [40–42]. This software is used to generalize the urban drainage system infrastructure into four modules including the atmosphere, surface water, groundwater, and transportation. Representing the study area in the model involves dividing the main basin into sub-basins and analyzing them separately to reach the conclusion of the study. To proceed with this modeling tool, the study area needs to be delimited first (Section 2.1) and to import the required maps. In the maps, the characteristics of sub-basins including the shape and length of connections, the characteristics of nodes, and the roughness coefficient need to be defined.

The statistics and the information from a meteorological synoptic station of the Ministry of Energy (Iran, Urmia) were used to prepare an urban runoff precipitation model. The network concentration time was found to be generally shorter than 2 h in most cities. Therefore, the duration of design rainfall usually did not exceed 3 h, and hence was considered for the calculations in the present study. Choosing 3 h when the concentration time was less than 2 h resulted in a low maximum precipitation rate using the rational method for peak flow.

There are generally three routing methods in the SWMM model including: (a) in constant current mode (steady); (b) kinematic wave; and (c) dynamic wave [43,44]. In this study, the dynamic wave routing mode was selected to simulate the runoff routing in the study area. There are three methods for the infiltration calculation in SWMM, including Horton's Method, the Green and Ampt, and the SCS curve number (CN) based on the calculation of cumulative infiltration by SCS method [45,46]. For the analysis of the present study, the SCS–CN method was selected.

There are also two main methods for calculating the head loss including the Darcy–Weisbach equation, and Hazen–Williams [47]. For this study, the Hazen–Williams equation was used to calculate the head loss in the study area (Equation (2)).

$$V = kCR^{0.63}S^{0.54} \tag{2}$$

In this equation, V is the velocity (m/s), k is a conversion factor for the unit system (0.849 for SI units), C stands for the roughness coefficient, R is the hydraulic radius (m), and finally S represents the slope of the energy line (head loss per length of pipe or hf/L) [48].

Then, sensitivity analysis was performed to evaluate the effects of the influencing parameters on the outcome of the model. Sensitivity analysis is a systematic method of evaluating the effects and presenting and justifying the results of calculated variables that result from changing the values of key parameters. The general definition of sensitivity is the rate of change in an output factor relative to the rate of change in an input factor [49]. In the present study, the effects of four influencing parameters including the slope, roughness coefficient, CN, and impermeability on the output of the model were evaluated.

### 2.4. Scenario Prioritization

A multi-criteria decision-making (MCDM) approach based on the simple additive weighting method (SAW) was finally implemented after discussing various scenarios based on the results achieved from the analysis performed in this study. SAW is among the most popular MCDMs proposed in 1981 by Hadang and Eun [50]. This method uses a linear increment function to represent the preferences of decision-makers. However, this technique is with the optimum performance when we assume that the preferences are independent or separate. According to this method, which is also known as the weighted linear combination method, after unscaling the decision matrix, using the weight coefficients of the criteria, the weighted unscaled decision matrix was obtained and the score of each option is calculated according to this matrix.

## 3. Results

### 3.1. Digital Watershed Modeling

In the present study, the initial step involved converting all the map data into information layers using Google Mapper (16.1). The output of the WMS software model has been illustrated in Figure 4, as described in 2.2. Furthermore, the waterway network, generated through the WSM model was converted into ArcGIS 13 features, as shown in Figure 5. Additionally, as depicted in Figures 6 and 7, the physiographic properties of the study basin were computed. Figure 7 also presents the output of Google Mapper. The inlet and outlet points of the sub-basins A, B, and C have been also indicated in Figures 8–10, respectively.

To calculate the watershed time of concentration, Equation (1) was utilized. The findings for basins A, B, and C are presented in Table 1. The time of concentration is an essential parameter in the hydrological analysis of a watershed, as it is the time required for runoff water to travel from the farthest point in the basin to the outlet. This calculation aids in understanding the behavior of rainfall–runoff processes, which helps in the design of effective stormwater management strategies. Providing the time required for runoff to reach the outlet (Equation (1)) contributes to better selection and sizing of various components of a stormwater management system, such as detention ponds, culverts, and other conveyance structures [51].

**Table 1.** The watershed time of concentration.

| | | |
|---|---|---|
| $t_{C(A)}$ | $0.366 \times 60 \times 1.67$ | 36.7 min |
| $t_{C(B)}$ | $0.344 \times 60 \times 1.67$ | 34.5 min |
| $t_{C(C)}$ | $0.408 \times 60 \times 1.67$ | 40.9 min |

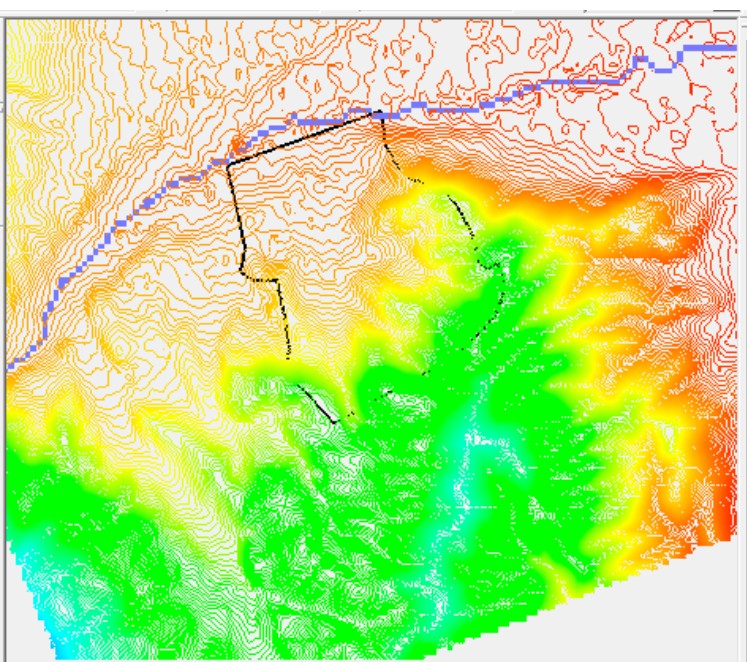

**Figure 4.** The output of the WMS software model.

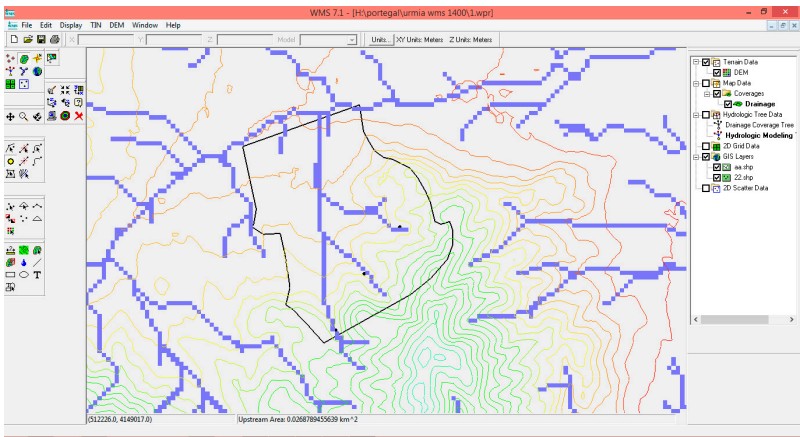

**Figure 5.** The schematic of the waterway network prepared using the WSM model in ArcGIS.

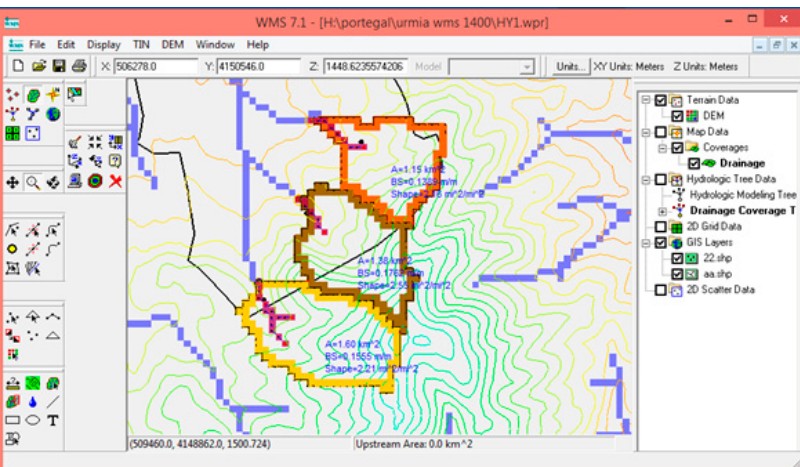

**Figure 6.** The physiographic properties of the study basin.

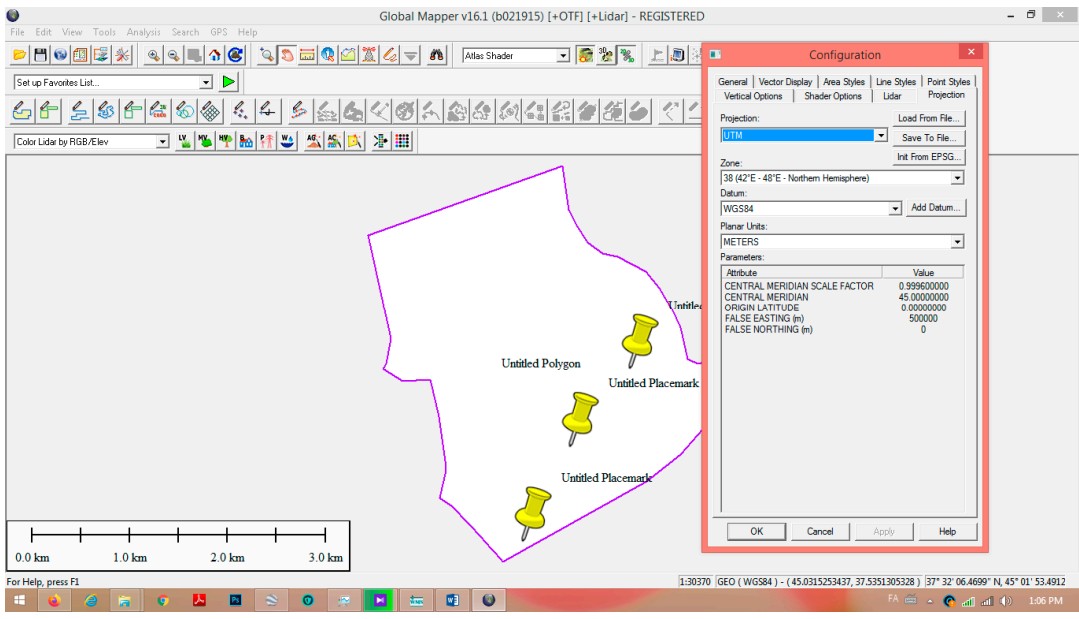

**Figure 7.** Output of Google Mapper.

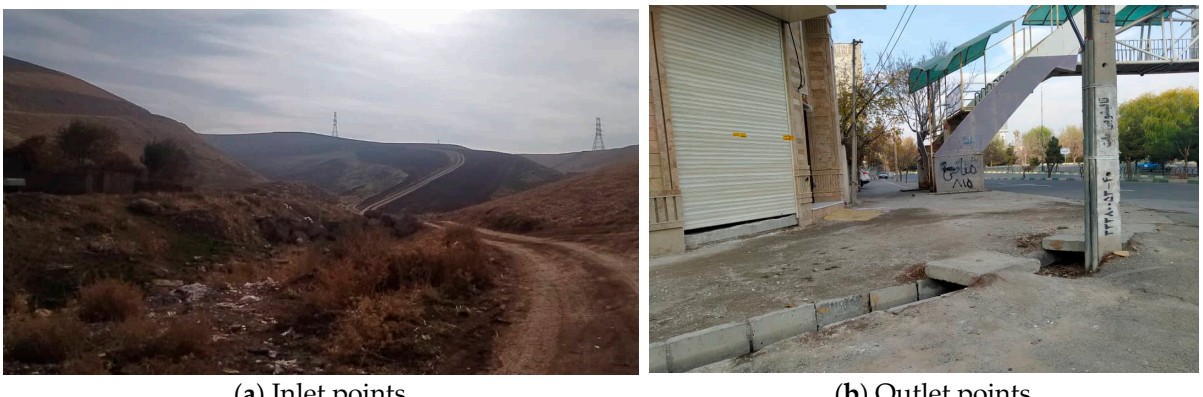

(**a**) Inlet points    (**b**) Outlet points

**Figure 8.** The inlet (**a**), and outlet (**b**) points of sub-basin A.

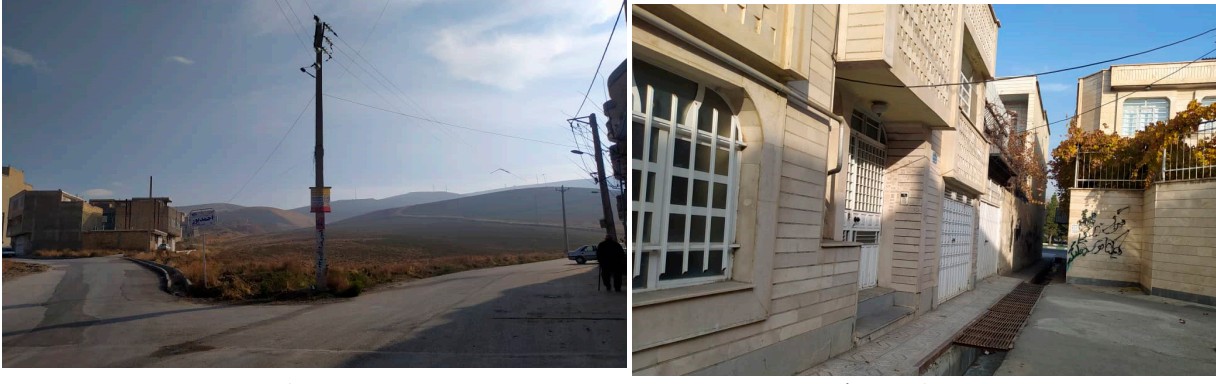

(**a**) Inlet points    (**b**) Outlet points

**Figure 9.** The inlet (**a**), and outlet (**b**) points of sub-basin B.

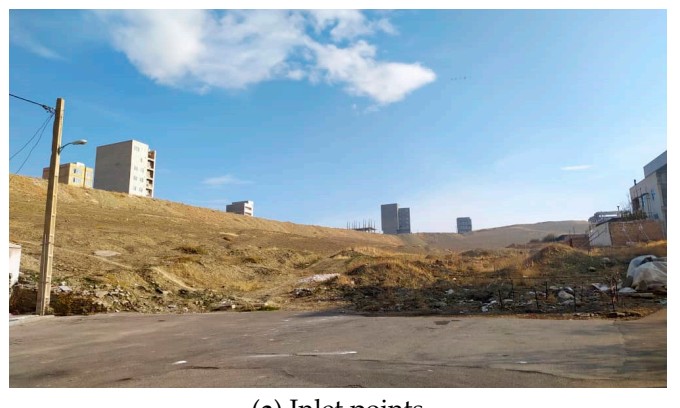

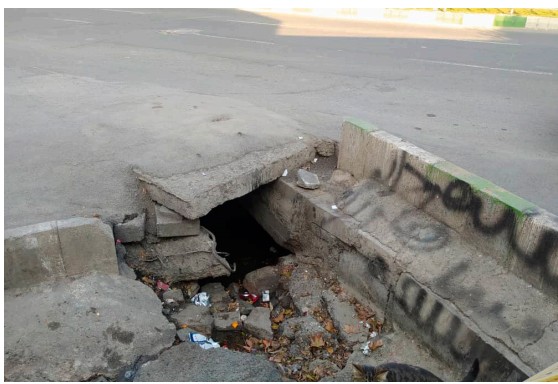

(**a**) Inlet points                                                                (**b**) Outlet points

**Figure 10.** The inlet (**a**), and outlet (**b**) points of sub-basin C.

This section provides a concise and precise description of the experimental results, their interpretation, as well as the experimental conclusions that can be drawn.

In the subsequent step of the study, the soil conservation service (SCS) artificial rainfall curve was generated for the return periods of 2 years, 5 years, and 10 years using stormwater management and design aid (SMADA) software, as a tool to assess stormwater runoff quantity and quality [52]. This software provides a platform for analyzing and modeling rainfall data to generate synthetic storms and is widely used in the field of hydrology. The SCS curve is a common method used for estimating the rainfall intensity and duration for different return periods and is utilized in the design of stormwater management structures [53]. The resulting curves for the 2-, 5-, and 10-year return periods are depicted in Figure 11, which provides a graphical representation of the rainfall flow versus time (m$^3$/s). These curves are used to estimate the expected rainfall amount for a given duration and return period, which is critical in the design of stormwater infrastructure to effectively manage runoff.

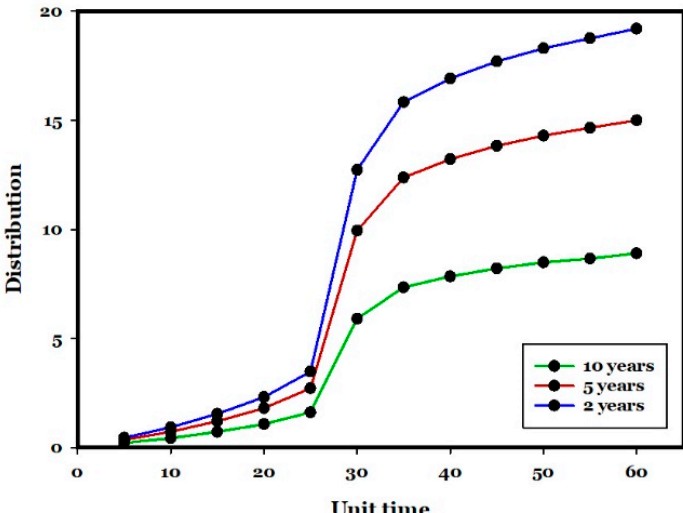

**Figure 11.** The soil conservation service (SCS) artificial rainfall curves with the return periods of 2 years (blue line), 5 years (red line), and 10 years (green line).

Then, the hydrograph of three sub-basins with return periods of 2, 5, and 10 years was calculated by the SCS method using WSM software as well as the rational method, as indicated in Figure 12a–c. The values are demonstrated in Table 2 (C = 0.14).

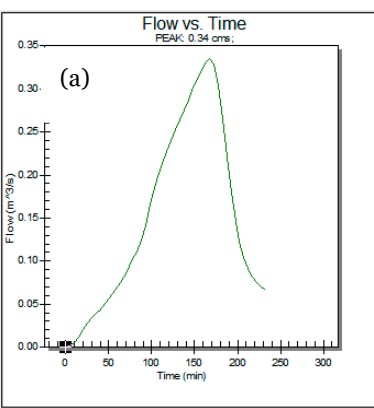
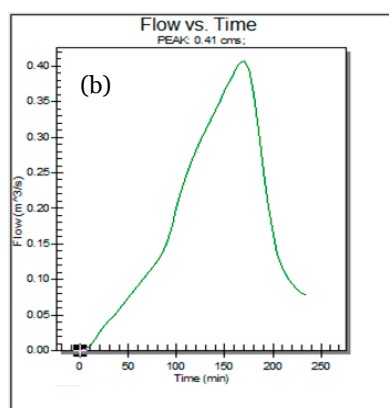
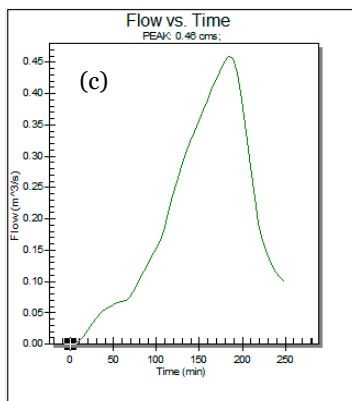

**Figure 12.** The hydrographs of sub-basins A (**a**), B (**b**), and C (**c**), with a return period of 2 years.

**Table 2.** Peak flow versus return period for the three sub-basins.

| Basin | Area (km²) | Flow (m³/s) (2 Years) | | Flow (m³/s) (5 Years) | | Flow (m³/s) (10 Years) | |
|---|---|---|---|---|---|---|---|
| | | SCs Model | Rational Method | SCS Model | Rational Method | SCs Model | Rational Method |
| A | 1.15 | 0.34 | 0.39 | 0.6 | 0.66 | 0.79 | 0.84 |
| B | 1.38 | 0.41 | 0.47 | 0.73 | 0.79 | 0.96 | 1 |
| C | 1.6 | 0.46 | 0.54 | 0.82 | 0.92 | 1.08 | 1.17 |

*3.2. SWMM Modeling*

To incorporate the sub-basins and surface water collection network components into the SWMM model, a schematic of the study area was first developed. This schematic served as a foundation for identifying the locations and characteristics of the sub-basins and connections within the network. The shape and length of the connections, the characteristics of the nodes, and the roughness coefficients were among the information introduced to the schematic and subsequently to the SWMM model. This step is crucial in the analysis of the hydrological processes, as it enables the simulation of the water movement within the network under various precipitation scenarios, which is essential in the design of effective stormwater management strategies. The surface water collection network includes 61 sub-basins, 141 junctions, 88 conduits, and 3 outlets. The surface water collection channel in the study area is in the form of an open rectangle with a floor width of 0.5–2 m and a depth of 0.5–1 m. The existing channel is made of concrete.

The results of the sensitivity analysis are presented in Table 3 including the subcatchment information regarding the slope, impervious fraction, roughness coefficient (pervious and impervious), and CN for the pervious area.

**Table 3.** The results of the sensitivity analysis.

| No. | Parameter | Changes | Changes in Output Peak Flow |
|---|---|---|---|
| 1 | Slope | +5% | No significant changes |
| 2 | CN | −5% | −4% |
| 3 | Roughness coefficient | +10% | −11% |
| 4 | Roughness coefficient | −10% | +13% |
| 5 | Impermeability | +10% | +15% |
| 6 | Impermeability | −10% | −19% |

Based on the results of sensitivity analysis, the impact of slope and curve number (CN) on the peak discharge hydrograph output was found to be relatively insignificant.

The highest sensitivity in the study area was observed for the percentage of impermeable surface, specifically due to the conversion from natural to residential land use. A 10% increase in impermeable surface resulted in a 15% increase in peak runoff hydrograph flow rate, while a 10% decrease in impermeability led to a 19% reduction in flow rate. The roughness coefficient exhibited significant sensitivity after impermeability, with a 10% increase causing an 11% decrease in peak flow rate in runoff hydrograph flow rate, and a 10% decrease causing a 13% increase in peak flow rate. Therefore, it is critical to carefully consider the selection of both the roughness coefficient and percentage of the impermeable surface when simulating urban runoff from an urban catchment.

Once the model was developed, it was verified and calibrated to ensure its accuracy. This involved testing the model's ability to perform adequately under known conditions, determined by measuring the flow rate at the outlets. The outcomes of this process are shown in Table 4. The purpose of this verification and calibration process is to confirm that the model can reliably simulate the behavior of the system under different scenarios and conditions. By demonstrating that the model works correctly, its credibility and accuracy can be established, and it can be used with greater confidence in future applications.

**Table 4.** SWMM model validation results were achieved based on the computational approach.

| Return Period (Years) | Flow Rate Calculated by the Rational Method (m³/s) | Calculated Flow Rate (m³/s) | Mean Relative Error |
|---|---|---|---|
| 2 | 16.85 | 16.22 | 3.7% |
| 5 | 28.4 | 29.8 | 4.7% |
| 10 | 36.35 | 39.8 | 8.7% |

The results of the model verification process indicate that the calculated error falls within an acceptable range. This proves that the SWMM model developed for simulating rainfall and surface runoff in Urmia, Iran, is reliable and accurate. The model's capability for simulating runoff hydrographs for different return periods is demonstrated in Figure 13 which depicts the runoff hydrographs for return periods of 2, 5, and 10 years across the entire study network. Additionally, Figure 14 presents the outflow diagrams for the three outlets for a 2-year return period. Upon examining outlet diagram 2, it is clear that the outlet was unable to fully discharge after 12 h, resulting in continued overflow for several hours.

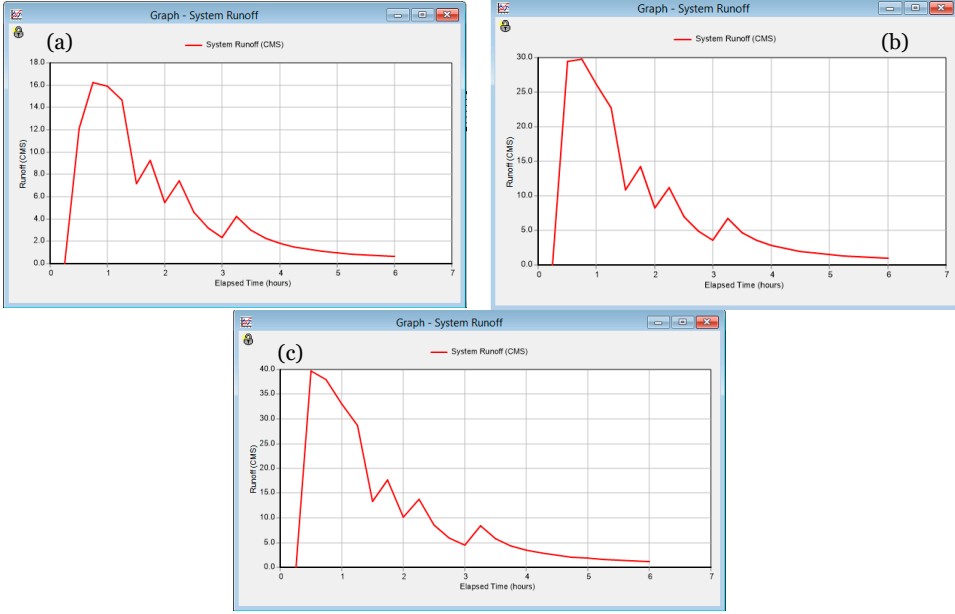

**Figure 13.** Hydrograph of the total network runoff output with 16.22 cubic meters per second and a return period of 2 years (**a**), with 29.8 cubic meters per second and a return period of 5 years (**b**), and with 39.8 cubic meters per second and a return period of 10 years (**c**).

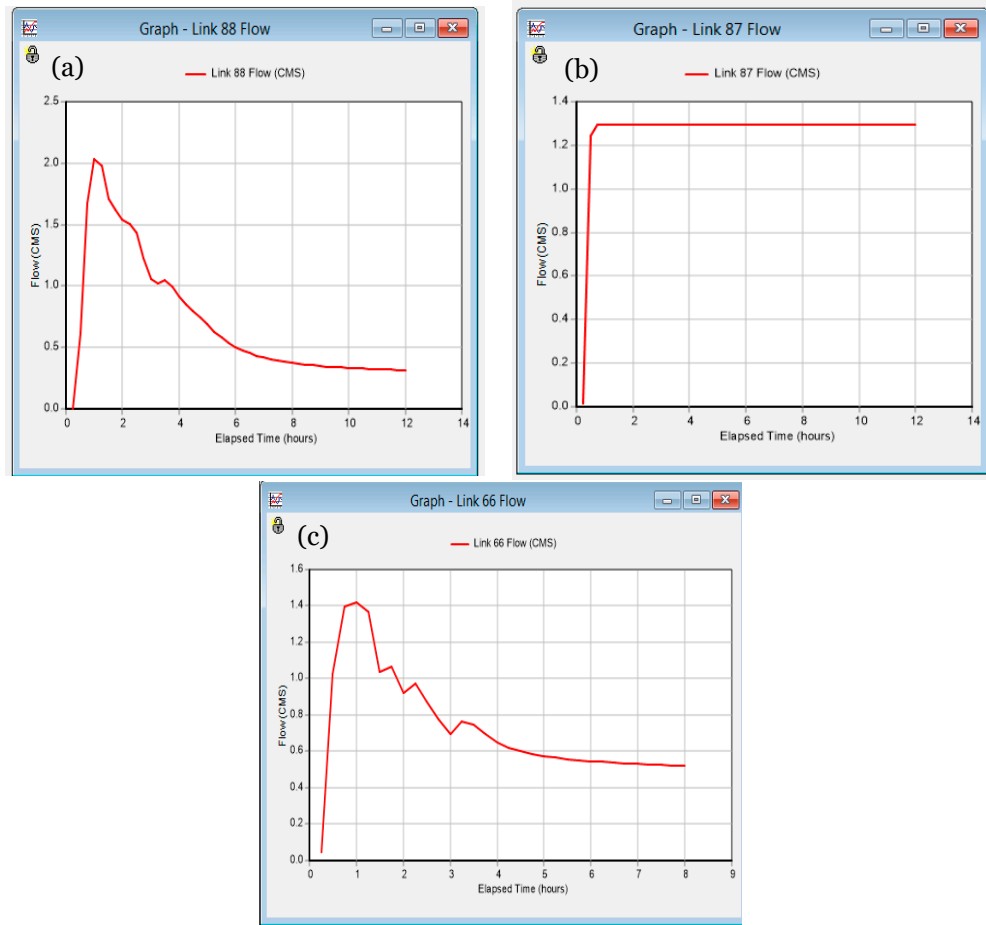

**Figure 14.** The outflow diagrams of the three outlets for a 2-year return period, ranging from 2.0 (**a**) × 1.3 (**b**) × 1.44 (**c**) cubic meters per second.

Moreover, the three outlets are only able to pass 30% of the peak flow for the 2-year return period, based on the peak flow value of 16.22 m³/s. These findings highlight the importance of carefully analyzing and considering the capacity of the drainage system's outlets in urban areas to prevent flooding during heavy rainfall events. The outflow diagrams of the three outlets for the 5-year and 10-year return periods are illustrated in Figures 9 and 10. According to outlet diagram 2, after 12 h the outlet was not able to fully discharge, and the overflow was ongoing for hours. According to the 5-year peak flow (29.8 m³/s), these three outlets could only pass 21% of the flow peak for the 5-year return period. According to Figure 13, outlet diagram A did not have a sharp peak, i.e., the channel was filled, which did not take more than 30 min, after which the complete discharge was completed. According to outlet diagram 2, after 12 h the outlet was not able to fully discharge, and the overflow was ongoing for hours. According to the 10-year peak flow (39.8 m³/s), these three outlets could only pass 17% of this flow peak for the 10-year return period.

To accurately assess the capacity of an existing drainage network using SWMM, it is crucial to evaluate various factors such as slope, speed, and height of runoff throughout the network. These factors play a crucial role in determining the network's ability to effectively manage stormwater runoff. Figure 15 provides a schematic representation of the network's channel slope, which is a critical factor that affects the surface flow discharge capacity for the rate of stormwater runoff. The steeper the channel slope, the faster the water will flow, which can increase the risk of flooding and erosion. Conversely, flatter slopes may lead to water accumulation and potential drainage system overload. Therefore, it is important to carefully analyze the slope of channels in the drainage network and other key factors to

ensure that the existing system is capable of managing stormwater effectively. By doing so, any necessary upgrades or modifications can be identified and implemented to improve the network's overall performance.

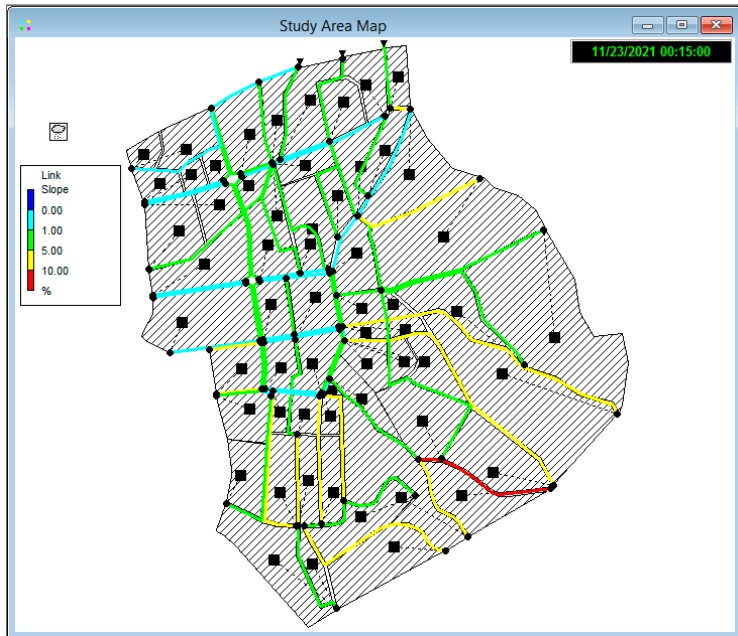

**Figure 15.** Slopes of the model links (ditches and pipes). The slope map illustrates the variation of the slope angles across the region, with each area color-coded to represent its corresponding slope angle.

As depicted in Figure 15, the southern region of the study area, situated in higher elevations, exhibits a steeper slope. This is a natural occurrence that is expected because of changes in the topography of the study area. Figures 16 and 17 supplement this observation by displaying schematic maps of the peak runoff heights in the existing channels and network junctions during a 2-year and 5-year return period, respectively. The maps presented in these figures are essential for analyzing the study area's drainage network performance during peak runoff periods. The green areas on the map represent overflow backflow zones that account for approximately 8.5% of the total network. Backflow overflow zones occur when the volume of water in the drainage system exceeds its capacity, resulting in water flowing over the bank of the ditch and flooding the basin surface in the opposite direction. Identifying such overflow backflow zones is crucial in understanding the drainage system's limitations and potential failure points during peak runoff events. This information can help in prioritizing maintenance and upgrades of the drainage network to ensure optimal performance and reduce the risk of flooding and other associated issues.

The present study also considers the speed of water flow in the channels of the network during different return periods, including a 2-year return period, which is represented in Figure 18, and a 5-year return period, which is represented in Figure 19. The total network speed during a 10-year return period is shown in Figure 20. Understanding the flow dynamics of a network of channels and nodes can be critical in predicting the potential for flooding and taking steps to mitigate flood risk. However, it is important to note that modeling water flow dynamics can be a complex and challenging task, particularly in urban areas with complex drainage systems and a variety of potential flood pathways. It is also important to consider the limitations of any modeling approach, including potential uncertainties and limitations in the data used to build the model.

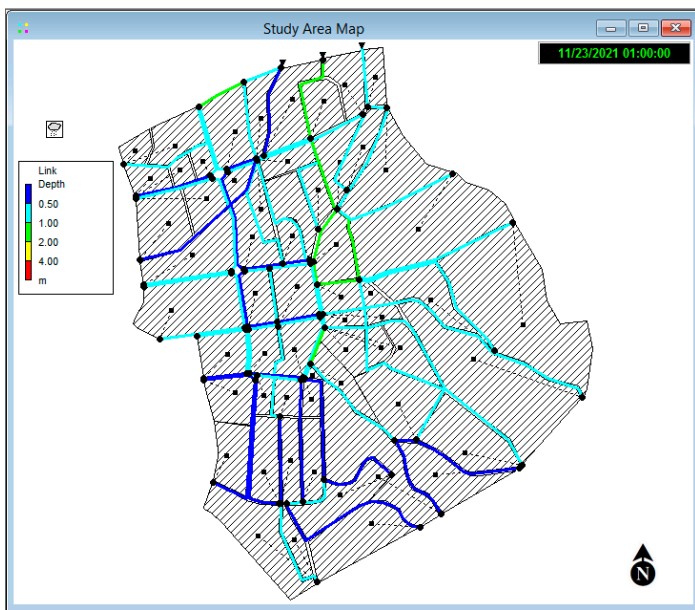

**Figure 16.** A schematic map illustrating the peak runoff height in the existing channels and network junctions during a 2-year return period.

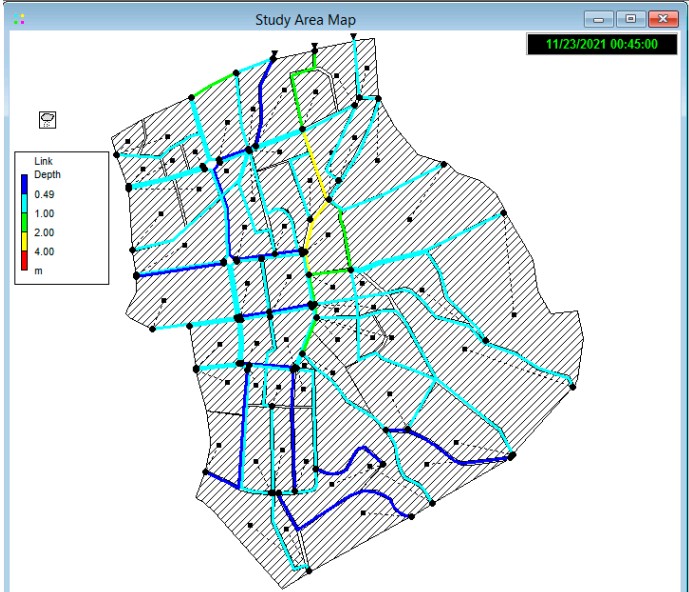

**Figure 17.** The runoff height at peak time in existing channels and nodes of the network for a 5-year return period.

The areas highlighted in yellow and red on the map represent parts of the network that are particularly vulnerable to flooding due to their high volume and flow of water. Based on the information presented in Figures 21–23, there is an urgent need to identify sustainable solutions to treat the areas represented in yellow and red on the map. Specifically, measures must be taken to reduce the volume and flow of water in these areas to prevent flooding and other related issues. To address this issue, sustainable and effective treatment methods must be identified and implemented in these areas. By reducing the volume and flow of water in these areas, the risk of flooding can be minimized, and the associated negative impacts such as damage to infrastructure, transportation disruptions, and loss of life can be avoided. This highlights the critical importance of effective water management strategies in

urban planning and the need for sustainable solutions that balance the competing demands of urban growth and environmental conservation.

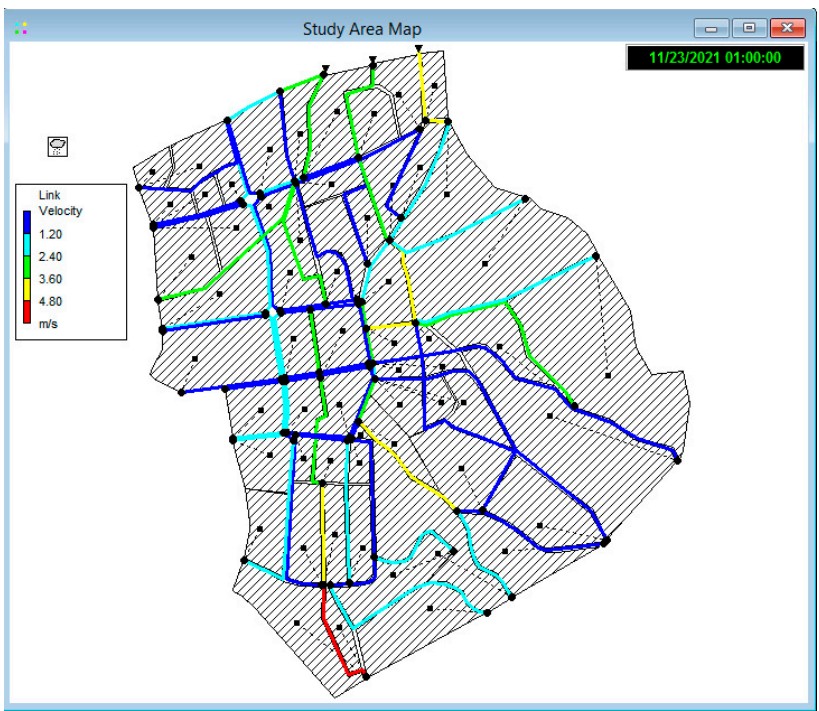

**Figure 18.** A schematic diagram of the speed (m/s) in the existing network channels for a 2-year return period.

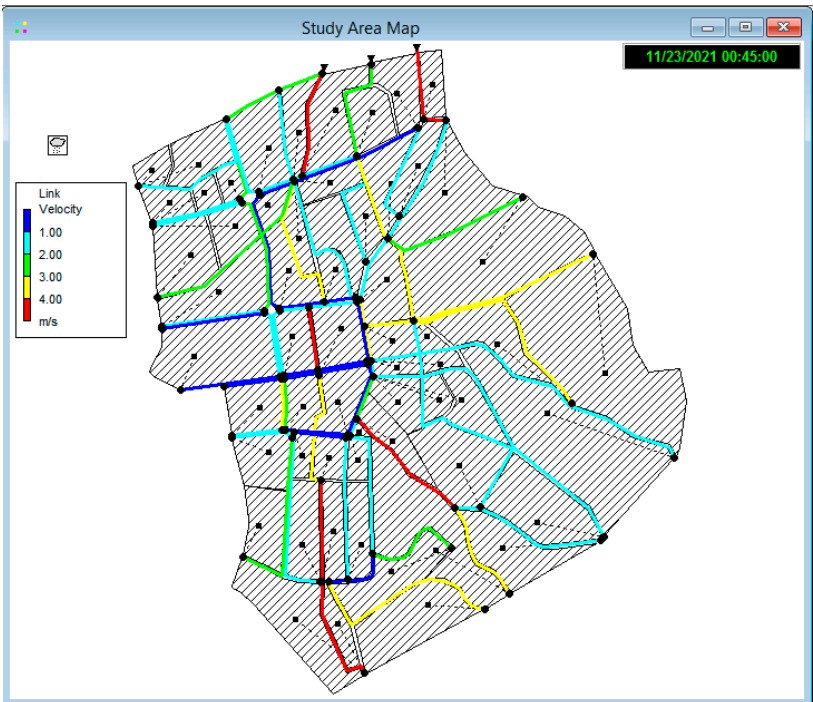

**Figure 19.** A schematic diagram of the maximum speed (m/s) for a 5-year return period.

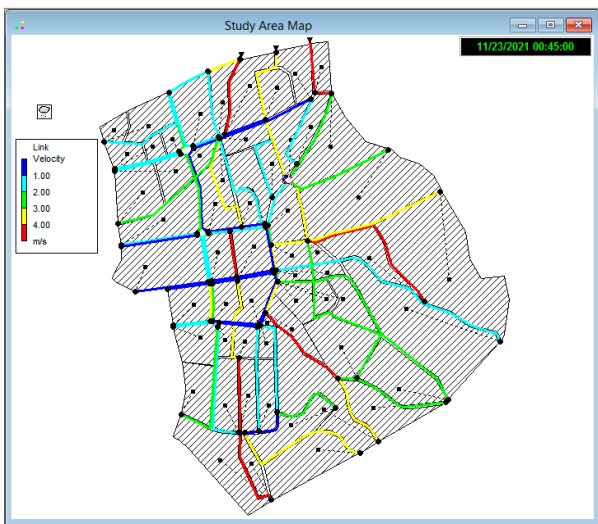

**Figure 20.** A schematic map of the total network speed for a 10-year return period.

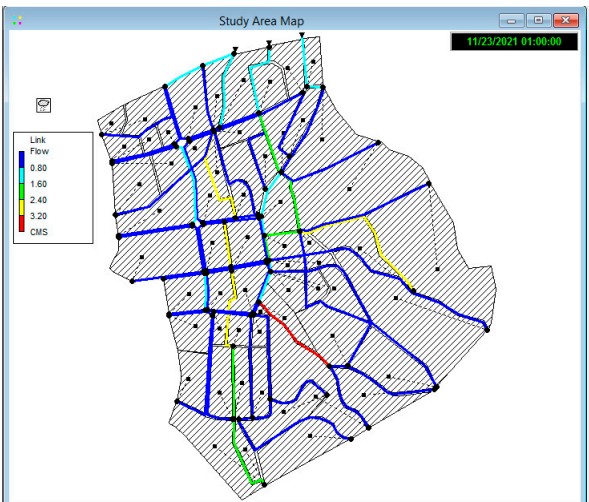

**Figure 21.** The transit flow map at 2-year peak time in the study area.

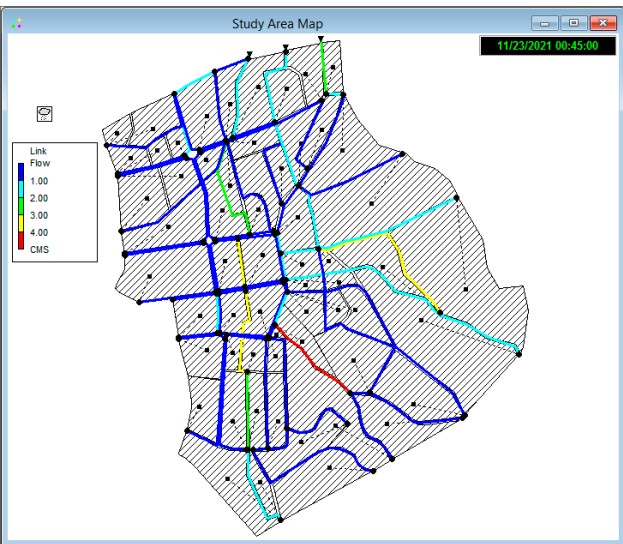

**Figure 22.** The transit flow map at 5-year peak time in the study area.

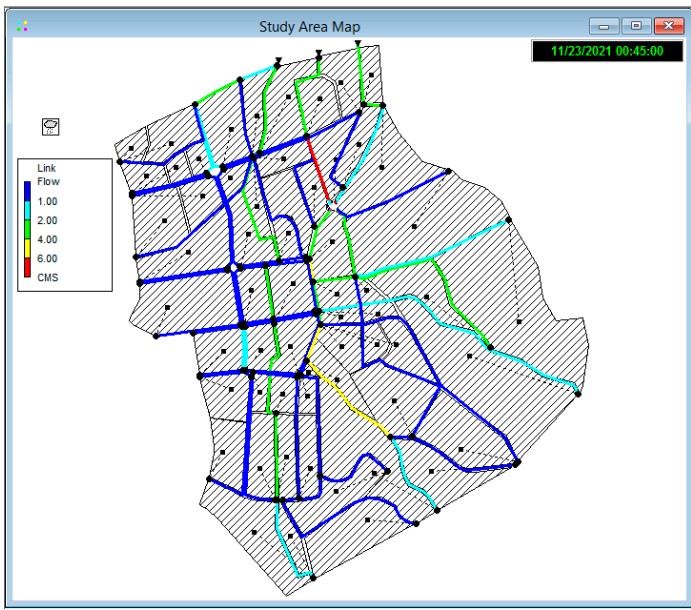

**Figure 23.** The transit flow map at 10-year peak time in the study area.

*3.3. Environmental Aspects*

The study conducted in Urmia City, Iran, highlights several key environmental aspects crucial for understanding and addressing stormwater management in urban areas. The environmental considerations are intricately tied to the hydrological processes and the urban infrastructure's ability to manage increased rainfall due to climate change.

The study's digital watershed modeling revealed critical insights into the hydrological dynamics of the region. By converting map data into information layers and analyzing the physiographic properties, the researchers identified how rainfall translates into runoff within various sub-basins. The time of concentration calculations, essential for understanding rainfall–runoff behavior, are crucial in designing stormwater management structures that can handle heavy rainfall without causing environmental degradation.

The sensitivity analysis indicated that impervious surfaces significantly affect peak runoff. Urbanization, characterized by the conversion of natural land to residential or commercial areas, increases impervious surfaces, leading to higher and faster runoff. This phenomenon not only increases the risk of flooding but also reduces groundwater recharge, impacting local water cycles and potentially leading to urban heat island effects. Effective stormwater management must thus address the balance between development and the preservation of permeable surfaces.

Using the soil conservation service (SCS) method and the stormwater management model (SWMM), the study assessed the quantity and quality of stormwater runoff for various return periods. The generation of artificial rainfall curves for 2-, 5-, and 10-year return periods helped in predicting the rainfall intensity and duration, which are critical for designing appropriate stormwater infrastructure. Properly managing stormwater quality is essential to prevent pollutants from urban areas from being washed into natural water bodies, thus protecting aquatic ecosystems.

Among the proposed solutions, vegetative swales were identified as the most promising. These swales can significantly enhance infiltration, reduce runoff velocity, and filter pollutants, providing a natural and sustainable method for stormwater management. The implementation of such green infrastructure not only mitigates flooding risks but also enhances biodiversity, improves air quality, and contributes to the aesthetic value of urban areas.

The study's detailed analysis of channel slopes and flow dynamics is crucial for understanding flood risks and designing effective stormwater systems. Steeper slopes in higher elevation areas increase runoff speed, which can exacerbate erosion and sediment

transport, negatively impacting downstream water quality and aquatic habitats. Conversely, flatter areas may suffer from water accumulation, stressing the importance of designing systems that consider local topographical variations.

### 3.4. Economic Aspects

The implementation of vegetative swales, green roofs, and rain gardens involves significant initial costs for construction, materials, and labor. Additionally, ongoing maintenance is required to ensure the effectiveness of these green infrastructure solutions, including regular cleaning, vegetation management, and repairs. Installing green infrastructure might also necessitate reallocating land or repurposing existing urban spaces, which could be challenging in densely populated areas.

Despite these costs, the benefits of such green infrastructure solutions are substantial. They effectively reduce peak flow rates during heavy rainfall events, minimizing flood risks and associated damage to infrastructure and property. Environmental benefits include improved water quality through natural filtration, enhanced groundwater recharge, and increased urban green spaces that contribute to biodiversity. Green roofs, for example, provide additional insulation for buildings, leading to reduced energy consumption for heating and cooling. The economic savings from reduced flood damage and lower maintenance costs for conventional drainage systems can be significant for municipalities in the long term. Furthermore, enhanced urban aesthetics and recreational spaces improve the quality of life for residents, promoting mental well-being and community engagement.

Public acceptance of the proposed solutions, including vegetative swales, green roofs, and rain gardens, is likely to be high given their numerous environmental, social, and aesthetic benefits. These green infrastructure solutions not only mitigate flood risks and enhance water quality but also improve urban aesthetics and create recreational spaces, contributing to a better quality of life. Public support can be further bolstered through community engagement and education about the long-term benefits of these solutions, such as reduced energy costs, increased biodiversity, and enhanced urban resilience to climate change. By involving local communities in the planning and implementation processes, the city can ensure that these initiatives meet public needs and preferences, thereby fostering a sense of ownership and responsibility towards sustainable urban development.

### 3.5. Social Aspects

The present study also highlights the significant social aspects. It was highlighted that effective stormwater management addresses environmental and economic concerns and has profound social implications that can be harnessed to enhance urban living conditions.

Community Health and Safety: If stormwater is managed properly, flood risks will be reduced, contributing to protecting residents from the hazards associated with flooding, such as property damage, waterborne diseases, and disruptions to daily life. Once flood risks are mitigated, the safety and well-being of inhabitants can be enhanced.

Urban Aesthetics and Recreational Spaces: The suggested vegetative swales, green roofs, and rain gardens as parts of green infrastructure enhance the visual appeal in Urmia and any other urban area. This is because these elements create more attractive, green, and open spaces, and are proved to be great areas for recreational activities for residents. The increased green spaces contribute to mental well-being, offering a natural environment for relaxation and social activities, which can strengthen community bonds.

Public Awareness and Education: It is essential to engage the community through educational programs and awareness campaigns that provide information about the profits of green infrastructure. This is a key aspect of social sustainability as it fosters a sense of environmental stewardship. City dwellers feel more committed to supporting and participating in sustainable urban development initiatives once they understand the long-term advantages of these solutions, including enhanced water quality, reduced flood risks, and improved urban resilience.

Community Involvement and Ownership: Another key area of concern is involving local communities in green infrastructure projects that aim at planning and implementation processes. This ensures that such initiatives adequately address public needs and community preferences.

## 4. Making Scenarios to Treat the Critical Conditions

The present study has identified three possible scenarios that could be implemented to address the issue of water flow in the urban medium. The first scenario involves modifying the existing stormwater network by increasing its length. This would involve installing runoff ditch raceways on both sides of the street, as currently some streets only have them on one side. This scenario aims to increase the network's capacity to drain more runoff from the area.

The second scenario proposed in the study involves adopting a low-impact development (LID) management strategy, such as using permeable pavements in the main streets. This approach aims to reduce the volume of runoff through surface adsorption. Previous studies have reported the effectiveness of permeable pavements in reducing runoff volume [54].

The third scenario proposed in the study involves implementing LID using vegetative swales. This approach aims to divert the runoff to green areas with trees along the street, which can help to reduce runoff volume and improve the aesthetic appeal of the area.

To evaluate the effectiveness of these scenarios, simulations were conducted to study their impact on water flow in the medium. The results of these simulations could provide valuable insights for policymakers and urban planners in developing effective water management strategies for the medium.

In summary, the study proposes three scenarios that could help address the issue of stormwater drainage flow in the urban medium, including modifying the existing network, adopting LID strategies such as permeable pavements, and implementing vegetative swales. By evaluating the impact of these scenarios through simulations, policymakers, and urban planners can identify the most effective strategies to manage water in the medium and promote sustainable urban development.

### 4.1. Scenario (a)

Scenario (a) involved building water ditches on both sides of the streets and enlarging them in certain areas to handle more runoff. The objective was to reduce flooding risk and associated issues like infrastructure damage and traffic delays.

Based on field visits, it was observed that the majority of streets either had no runoff ditch raceway to drain the water or had only one side (scenario 1). This led to a significant volume of water accumulating on the streets during rainfall, which could result in flooding, traffic delays, and other related issues. To address this issue, the study proposes constructing water ditch raceways on both sides of the street and increasing the dimensions in certain areas to increase the capacity to drain a larger volume of runoff from the study area. This approach aims to reduce the risk of flooding and minimize the associated negative impacts such as damage to infrastructure and transportation disruptions. Figure 24 represents the runoff hydrograph resulting from simulating scenario (a), which involves modifying the existing network by constructing water raceways on both sides of the streets. The simulation results demonstrate the potential effectiveness of this approach in reducing the volume of runoff in the study area and mitigating the risk of flooding.

Implementing scenario (a) would require significant investment in infrastructure, and its effectiveness may depend on several factors such as the slope of the streets and the soil type. Therefore, further analysis and evaluations are necessary to determine the feasibility and potential effectiveness of this approach in the context of the study area.

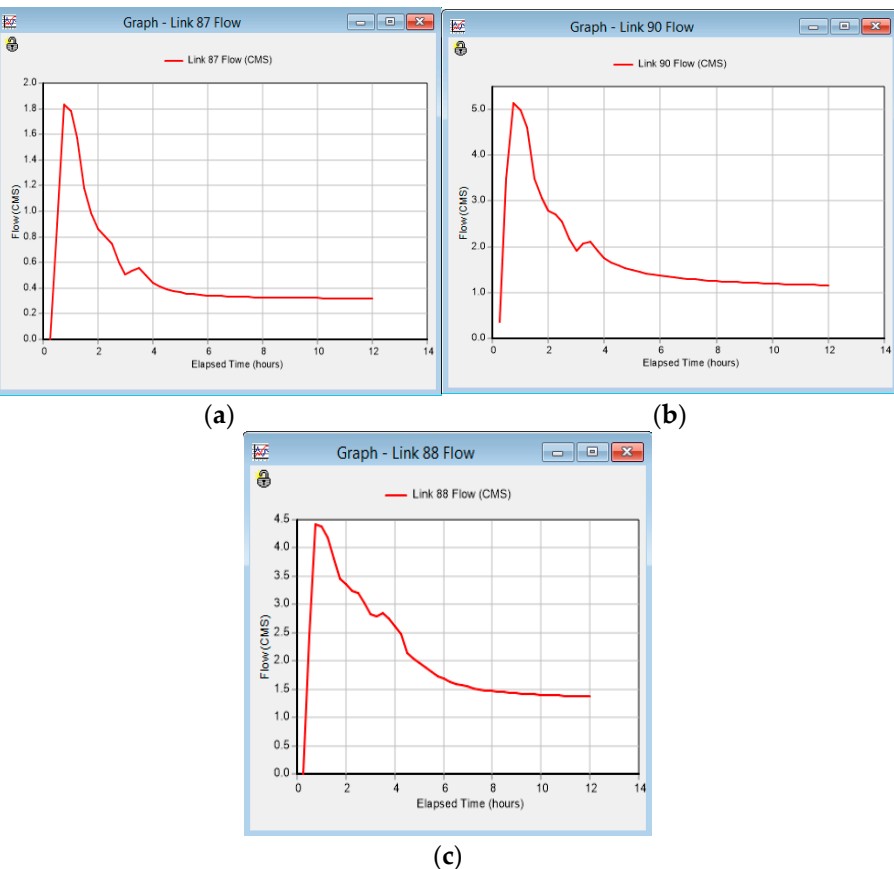

**Figure 24.** Flow rate for a 10-year return period ranges from 1 to 3—5.14 (**a**) × 1.83 (**b**) × 4.41 (**c**) cubic meters per second, respectively, as a result of the implementation of scenario a.

Based on the information presented in Figure 24, it appears that the implementation of scenario (a) did not result in any flow retraction issues in the study area. Figure 25 represents the transition flow map at the 10-year peak time following the implementation of scenario (a). The figure suggests that there are no observable yellow and red regions that could be attributed to the implementation of this scenario.

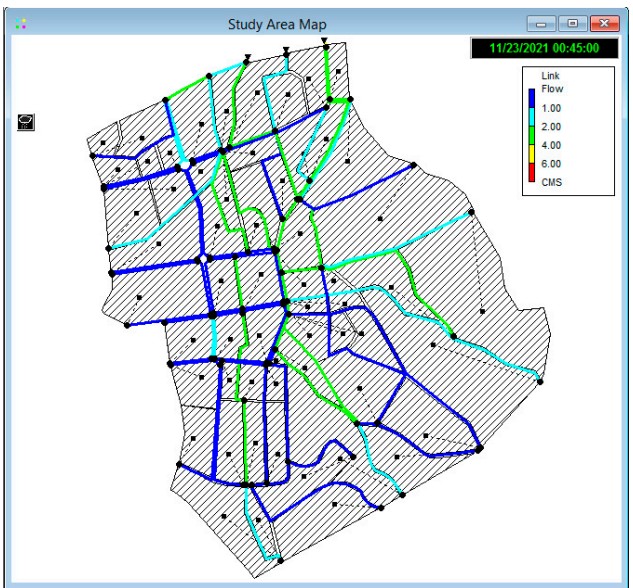

**Figure 25.** Transition flow map at 10-year peak time, as a result of the implementation of scenario a.

### 4.2. Scenario (b)

Scenario (b) takes advantage of the surface infiltration strategy to ensure that some runoff water is kept out of the drainage network. It relies on using different permeable pavements that help eliminate issues related to drainage system overloads.

For a considerable period, countries like Iran have implemented a management strategy known as low-impact development, which primarily relies on surface absorption, also known as surface infiltration. This strategy ensures that a portion of the runoff water is prevented from entering the drainage network. Figure 23 represents various forms of permeable concrete pavements. The effects of adopting such strategies can be investigated using SWMM software version 5.2.4. Figure 26 indicates the runoff hydrograph resulting from simulating the scenario (b). By examining the runoff hydrograph in this figure, it is possible to gain insights into the effectiveness of scenario (b) as a management or policy approach. This information can be used to inform future decision-making and improve water management strategies in the study area and beyond.

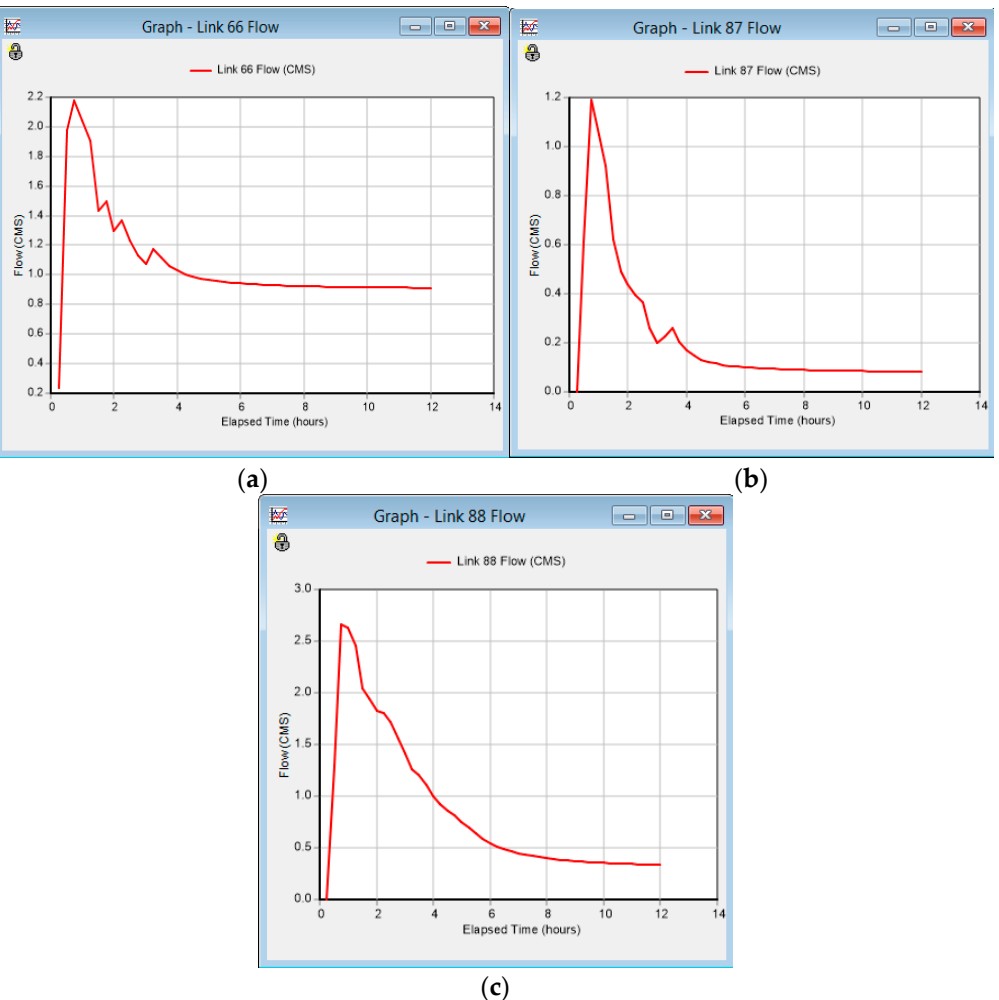

**Figure 26.** Flow rate for a 10-year return period ranges from 1 to 3—2.2 (**a**) × 1.2 (**b**)× 2.66 (**c**) cubic meters per second, respectively, as a result of the implementation of scenario b.

According to this figure, using permeable pavements results in the elimination of the issues related to drainage system overloads. In other words, flow retraction could be performed efficiently as the peak flow rate of the hydrograph was reduced. In addition, no critical point could be observed in the transition flow map at a 10-year peak time after implementation of scenario b, as illustrated in Figure 27.

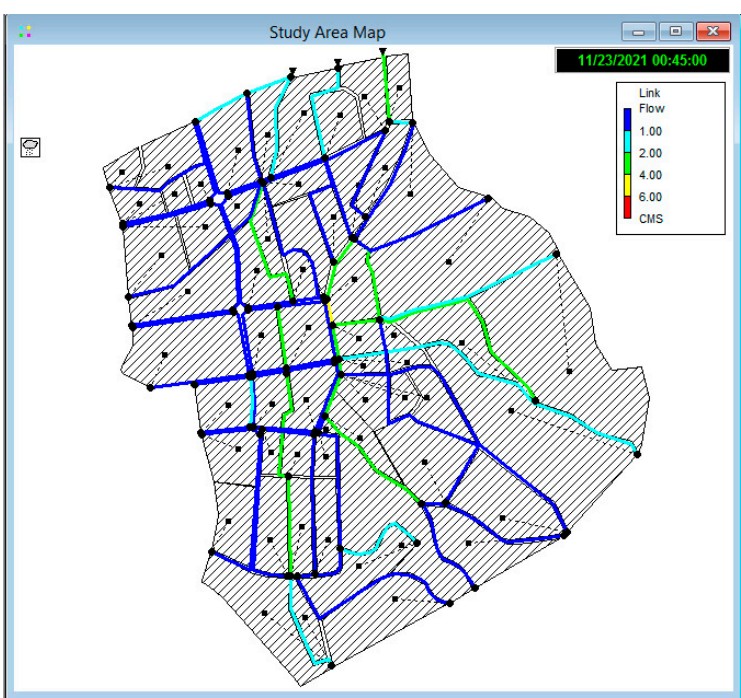

**Figure 27.** Transition flow map at 10-year peak time, as a result of the implementation of scenario b.

*4.3. Scenario (c)*

Scenario (c) integrates green infrastructure into the water management system. It uses vegetative swales and is expected to reduce runoff and also supports the greenery of urban areas while enhancing surface infiltration.

As mentioned before, this method has been widely used in many countries in order to reduce runoff by surface adsorption (surface infiltration). In this method, the path combines water and vegetative swale (green space along the streets) (Figure 28). Figure 29 indicates the runoff hydrograph resulting from simulating the scenario c.

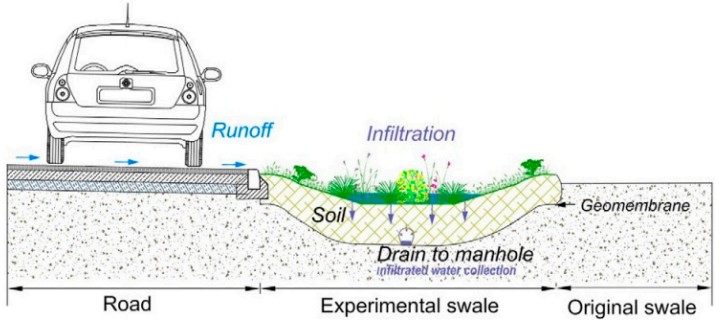 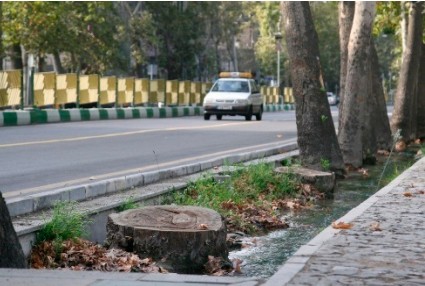

**Figure 28.** A schematic of the vegetative swale principals, adopted from Leroy et al., (2016) (**left**) [53], a real application of such a strategy, Valiasr St., Tehran ((**right**), picture by the author).

According to Figure 19, the implementation of vegetative swale resulted in overcoming the issues related to the rainfall flow, as the peak flow rate of the hydrograph reduced (Figure 30).

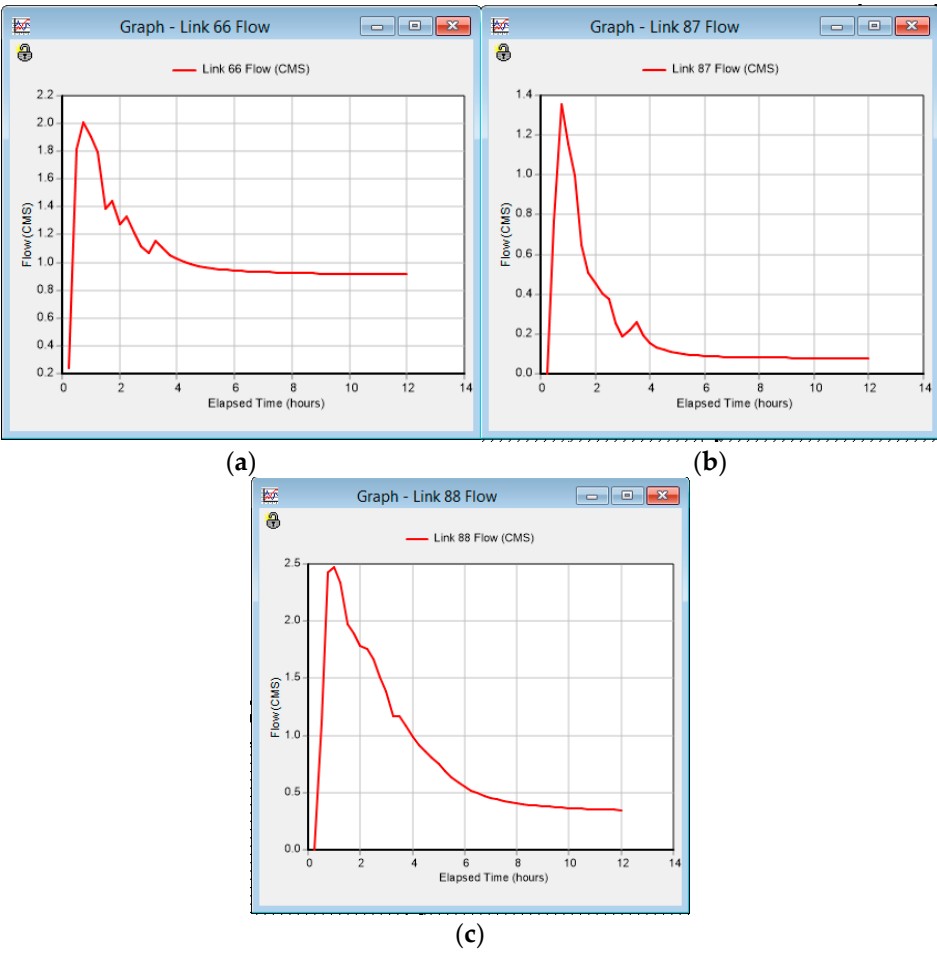

**Figure 29.** The flow rate for a 10-year return period ranges from 1 to 3—2 (**a**) × 1.36 (**b**) × 2.48 (**c**) cubic meters per second, respectively, as the result of the implementation of scenario c.

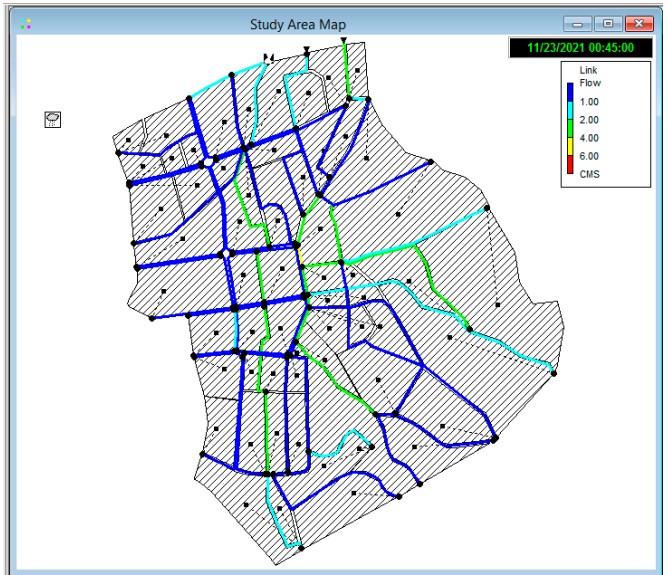

**Figure 30.** Transition flow map at 10-year peak time, as a result of the implementation of scenario c.

### 4.4. Scenario Prioritization

To identify criteria that could affect the management of heavy rainfall design issues and prioritize scenarios, a decision matrix was established in this study. Such decision-making

matrices have been already used to deal with a wide range of issues such as urban traffic management [55], and selection of the new sites for solid waste management [56]. After identifying the criteria and making the table, the decision matrix was unscaled through a process called unscaling the decision matrix. If a criterion was positive, each number in that column was divided by the largest number, and if a criterion was negative, the minimum of that column was divided by each number. Next, a weighted matrix was established based on weights calculated from other methods. To select the best option, the rows of the weighted matrix were added, and the score of each option was calculated. The options were then ranked based on their score, and the total score of the criteria was expected to be equal to 1. Table 5 lists the seven criteria with different weights that were defined to prioritize according to the objectives of the project. These criteria were selected and prioritized in the present study based on the simple additive weighting method (SAW) method.

**Table 5.** The 7 criteria with varying weights, which were established to prioritize based on the main objective of the study.

| Identification | Definition | Value |
|---|---|---|
| C1 | Purpose of the study (minimum traffic at runtime) | 0.15 |
| C2 | Implementation time | 0.05 |
| C3 | Lowest implementation cost | 0.15 |
| C4 | Reduced flooding and solve the problem | 0.25 |
| C5 | Minimum implementation space limit | 0.2 |
| C6 | Least environmental issues | 0.1 |
| C7 | Social implementation tensions | 0.1 |

Tables 6 and 7 present the interval scores and the final prioritization of the measures based on the scenarios defined in order to manage the issues related to the rainfall flow in the study area, respectively. According to these tables, the priority of the actions are as follows:

(a)   The vegetative swale (the third scenario);
(b)   The second priority is the first scenario, i.e., to improve the network;
(c)   The third priority is the second scenario, i.e., permeable pavement.

**Table 6.** The interval scores of various scenarios studied in the present manuscript are based on the criteria listed in Table 5.

| C1 | | C2 | | C3 | | C4 | | C5 | | C6 | | C7 | |
|---|---|---|---|---|---|---|---|---|---|---|---|---|---|
| R1 | 0.15 | R1 | 0.05 | R1 | 0.15 | R1 | 0.25 | R1 | 0.20 | R1 | 0.10 | R1 | 0.10 |
| R2 | 0.30 | R2 | 0.10 | R2 | 0.30 | R2 | 0.50 | R2 | 0.40 | R2 | 0.20 | R2 | 0.20 |
| R3 | 0.45 | R3 | 0.15 | R3 | 0.45 | R3 | 0.75 | R3 | 0.60 | R3 | 0.30 | R3 | 0.30 |

**Table 7.** The interval scores of various scenarios studied in the present manuscript are based on the criteria listed in Table 6.

| Criteria | C1 | | C2 | | C3 | | C4 | | C5 | | C6 | | C7 | | |
|---|---|---|---|---|---|---|---|---|---|---|---|---|---|---|---|
| Weights | 0.15 | | 0.05 | | 0.15 | | 0.25 | | 0.20 | | 0.10 | | 0.1 | | Sum |
| Values | Value | Sum | Value | Sum | Value | Sum | Value | Sum | Value | Sum | Value | Sum | Value | Sum | |
| S1 | 1.00 | 0.15 | 1.00 | 0.05 | 2.00 | 0.30 | 2.00 | 0.50 | 3.00 | 0.60 | 2.00 | 0.20 | 1.00 | 0.10 | 1.90 |
| S2 | 2.00 | 0.30 | 2.00 | 0.10 | 1.00 | 0.15 | 1.00 | 0.25 | 1.00 | 0.20 | 1.00 | 0.10 | 2.00 | 0.20 | 1.30 |
| S3 | 3.00 | 0.45 | 3.00 | 0.15 | 3.00 | 0.45 | 3.00 | 0.75 | 2.00 | 0.40 | 3.00 | 0.30 | 3.00 | 0.30 | 2.80 |
| Ranking | R1 (Improve) | | | | R2 (Permeable Pavement) | | | | | R3 (Vegetative Swale) | | | | | |
| | 2 | | | | 3 | | | | | 1 | | | | | |

*4.5. Scenario Analysis*

Scenario (a) involves constructing water ditches on both sides of the streets and enlarging them in certain areas to handle more runoff, aiming to reduce flooding risk and associated infrastructure damage and traffic delays. While this approach is effective in mitigating runoff, it requires significant infrastructure investment and may be influenced by factors such as street slope and soil type. The long-term effects of this scenario include a reduced risk of urban flooding and less damage to infrastructure, but the ongoing maintenance costs and the potential need for further modifications in the future could be substantial. Additionally, the environmental impact is relatively neutral as it does not significantly enhance urban green spaces or biodiversity.

Scenario (b) employs permeable pavements to enhance surface infiltration, reducing the volume of runoff entering the drainage network. This method helps prevent drainage system overloads and is effective in reducing peak flow rates but may involve considerable costs for materials and installation. The long-term benefits of this scenario include improved groundwater recharge and reduced surface runoff, which can alleviate pressure on urban drainage systems. However, the economic costs can be high due to the need for specialized materials and regular maintenance to ensure the permeability of the pavements. Environmentally, this approach offers some benefits in terms of reduced runoff and potential cooling effects in urban areas, but it doesn't provide the same level of green space or habitat enhancement as other green infrastructure solutions.

Scenario (c) integrates green infrastructure, specifically vegetative swales, to reduce runoff and support urban greenery. This approach not only reduces peak flow rates but also enhances environmental benefits such as increased biodiversity, urban cooling, and improved air quality. The long-term effects include a more resilient urban environment capable of handling heavy rainfall events, along with the added aesthetic and recreational benefits of green spaces. Economically, while the initial costs of implementation might be significant, the long-term savings from reduced flood damage and lower maintenance costs for traditional drainage systems can be substantial. Social acceptance of vegetative swales is generally high, as they improve the quality of life by providing attractive green spaces and recreational areas for residents. The environmental impacts are overwhelmingly positive, as vegetative swales enhance biodiversity, promote sustainable water management, and contribute to urban cooling.

Among the three scenarios, scenario (c) stands out as the best option due to its multifaceted benefits. It effectively manages runoff, offers significant environmental advantages, and enjoys high social acceptance due to its aesthetic and recreational value. The long-term environmental and social benefits of integrating green infrastructure outweigh the initial economic costs, making scenario (c) the most sustainable and effective solution. While scenarios (a) and (b) provide important benefits, they do not offer the same comprehensive improvements to urban resilience, environmental health, and community well-being as scenario (c). Therefore, scenario (c) represents the optimal strategy for addressing stormwater management challenges in Urmia City, providing a robust framework for sustainable urban development.

## 5. Discussion

The paper performed a comprehensive analysis of the hydrological processes in Urmia, Iran, and employed both digital watershed modeling and Storm Water Management Model (SWMM) modeling techniques for this purpose. The results of sensitivity analysis highlighted the significant impact of impermeable surface percentage and roughness coefficient on peak flow rates, which indicates the importance of model parameterization in this regard. In addition, results of water movement under various precipitation scenarios through integrating sub-basins and surface water collection network components into the SWMM model were reliable and accurate, showing that it was effective in simulating runoff hydrographs for different return periods. The study also found that scenario (c), which proposed the development of a vegetative swale, was the most optimal alternative for

reducing peak flow rates as it mitigated the overload of the drainage system and offered efficient flow management. The results emphasized the significance of analyzing drainage system outlets' capacity to thwart flooding events during heavy rainfall events in the study area. It was observed that outlets did not sufficiently discharge under peak flow conditions, and this highlights the importance of careful evaluation and potential upgrades of drainage systems in the area.

However, incorporating additional green infrastructure solutions, such as green roofs and rain gardens, could enhance stormwater management. Green roofs can reduce runoff volume and delay flow into the stormwater system while providing thermal insulation and mitigating urban heat island effects. Rain gardens can capture and infiltrate stormwater runoff from impervious surfaces, enhancing groundwater recharge and improving water quality. Integrating these solutions with vegetative swales creates a multifaceted strategy, maximizing the environmental, social, and economic benefits of urban green infrastructure.

## 6. Conclusions

Heavy rainfall can cause significant damage to the urban infrastructure and the environment, including flooding and landslides. Countries around the world have implemented various strategies to mitigate the impact of heavy rainfalls, based on their geographic location, climate conditions, and available resources. This study used a combination of WMS and SWMM to identify areas in Urmia City that require stormwater management interventions and recommend effective solutions to mitigate the negative effects of heavy rainfall events. By identifying vulnerable points and selecting and discussing possible scenarios to manage flooding under heavy rainfall, the study provided valuable insights into the most effective strategies for managing the issue. The establishment of a decision matrix through the prioritization of influential criteria using a simple additive weighting method (SAW) provided a clear and objective approach to selecting the most effective scenario. The third scenario, which involved the application of a vegetative swale, was identified as the most promising solution to deal with the issue. As a conclusion, the study's results provide a valuable framework for decision-makers in Urmia City, as well as other areas facing similar issues, to effectively manage heavy rainfall events and mitigate their negative effects.

In the context of this study in Urmia, Iran, climate change significantly impacts urban drainage systems by altering precipitation patterns and intensifying rainfall events. This results in increased runoff and heightened flood risks, particularly in urban areas with limited natural drainage capacity due to extensive impervious surfaces. The adaptability of drainage systems in the long term involves implementing sustainable strategies such as green infrastructure (e.g., vegetative swales, green roofs) to enhance water absorption and reduce runoff volumes. These solutions not only mitigate flood hazards but also offer environmental benefits like improving water quality and supporting urban biodiversity. Additionally, incorporating advanced modeling techniques and adaptive management practices can optimize the resilience of drainage systems to cope with future climate uncertainties. Collaborative efforts among stakeholders, including local communities and authorities, are essential to ensure the effectiveness and sustainability of these adaptive measures in addressing ongoing and future challenges posed by climate change on urban drainage systems in Urmia and similar urban settings.

**Author Contributions:** Conceptualization: R.M.A., J.F.S., and J.M.; methodology: R.M.A., J.F.S., and J.M.; software, R.M.A.; investigation, R.M.A.; resources, J.F.S. and J.M.; data curation, R.M.A.; writing—original draft preparation, R.M.A.; writing—review and editing, R.M.A., J.F.S., and J.M.; visualization, R.M.A., J.F.S., and J.M.; supervision, J.F.S. and J.M.; project administration, J.F.S. and J.M. All authors have read and agreed to the published version of the manuscript.

**Funding:** The authors acknowledge the financial support by FCT—Fundação para a Ciência e Tecnologia, I.P. by project reference UIDB/04450/2020 and DOI identifier 10.54499/UIDB/04450/2020.

**Institutional Review Board Statement:** Not applicable.

**Informed Consent Statement:** Not applicable.

**Data Availability Statement:** Data are contained within the article.

**Conflicts of Interest:** The authors declare no conflicts of interest.

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
