# Peer review of "Designing Sustainable Drainage Systems as a Tool to Deal with Heavy Rainfall—Case Study of Urmia City, Iran"

_sustainability, doi:10.3390/su16177349_

Round 1

Reviewer 1 Report

Comments and Suggestions for Authors

This manuscript (sustainability-2986632) tries to employ a combination of the Watershed Modelling System (WMS) and Stormwater Management Model (SWMM) to identify areas in Urmia City, Iran that require storm water management to develop a comprehensive understanding of the hydrological processes within the study area and to prevent the subsequent effects of heavy rainfall. Although the study fits the aim and scope of this journal, its novelty and contribution should be highlighted more clearly. Another concern is that some related latest studies have been neglected. Thus, a major revision is necessary. More detailed suggestions and comments are presented as follows:

- . It is very uncommon that the authors directly mentioned what they did and the conditions of the study area in the very first part of the Abstract. Most contents in the Abstract are related to this study area only. Therefore, some international implications are needed in this part.

- . From Line 71 to 73, the authors mentioned that: "Among all the mentioned strategies developed to control heavy rainfalls, the implementation of sustainable drainage systems can bring advantages over other methods due to their ability to provide rapid and efficient removal of storm water runoff". This statement is too subjective because some other methods are also effective, such as the redesign of urban land use.

- . In the Introduction Section, the authors should state the novelty of the study and the specific research questions or hypotheses. A well-defined problem statement will help readers understand the significance of your work and its contribution to the field. In the current manuscript, some related studies have been mentioned but without a detailed summary.

- . I am missing the hydrological conditions of the study areas, which are very important to this study.

- . In particular, the literature review section is not even enough. There are a number of studies that have considered land use in flood mitigation. This aspect has been ignored in the manuscript. Please see below for example.

Assessing the scale effect of urban vertical patterns on urban waterlogging: An empirical study in Shenzhen, 2024, 107486.

From risk control to resilience: developments and trends of urban roads designed as surface flood passages to cope with extreme storms. Front. Environ. Sci. Eng. 18, 22

- . The parameter settings of the Watershed Modelling System and SWMM model, the specific process of data input, and the method of model verification were not described in detail.

- . There is a lack of detailed description and evaluation of the sources, collection methods, accuracy and completeness of data on hydrology, topography, and meteorology.

- . The authors should not just post the screenshots of the software in the manuscript. The results should be displayed in a much clearer manner.

- . Almost all the geographical maps lack the legends, scale bars, and compass.

- . There is also insufficient comparison and discussion of the simulation results with the actual situation, which makes it difficult for readers to evaluate the practicality of the simulation results.

- . This manuscript is too simple in terms of the Conclusion Section, which does not fully summarize the take home messages and contributions of the study, and does not clearly point out the limitations of the study and possible future research directions.

- . In addition, many of the figures should be improved in terms of their visual aspect.

Comments on the Quality of English Language

Moderate editing of English language required.

Author Response

I sincerely appreciate time and patience dear Reviewers dedicated to reading my manuscript. Their comments and constructive feedback significantly contributed to enhancing the quality of my research. The following modification have been made based on their comments and highlighted green in the manuscript.

Reviewer 2 Report

Comments and Suggestions for Authors

The article deals with current and important issues and fits in with the theme of the journal. However, several problems should be pointed out that reduce the value of the study and should be corrected:

- Chapter Materials and Methods - lines 109-113 - unclear information without access to the indicated drawings.

- Subsection "2.3 Modelling with stormwater management model (SWMM)" - lines 170-171 "There are also two main methods for calculating the head loss including Darcy-the Weisbach equation, and Hazen Williams. Should be "the Darcy-Weisbach equation." The sentence is true only for channels with a "force-main" cross-section. In other cases, Manning's formula is used.

- Subsection "2.4 Scenario prioritization" - lines 199-213 - the content of the template file was left.

- Subsection "3.1 Digital watershed modeling" and Figure 1: "2 years, and 5 years" or maybe "2, 5, and 10 years" - the information is not consistent.

- Subsection "3.2 SWMM modelling" - sensitivity analysis omits Width parameter.

- Table 4: Only the maximum flow rate values were evaluated?

- Figures 3, 4 were probably done with a Time step of 15 minutes; Figures 14, 16 and 19 Time step reduced to probably a value of 10 minutes. This should be standardized and the Time step set to at most 5 minutes.

- If Speed and Flow parameters (Figures 8-13) are presented for 2, 5 and 10 years, why is Depth only for 2 and 5 years? One could consider presenting Max Flow and Max Velocity values instead of values for the selected time and instead of Depth show Max Capacity (SWMM 5.2). If there is flooding, one could consider including graphics with visualization of the Flooding parameter.

- The conclusions are correct (limitation of runoff by the sewer system), indicate whether similar ones have already been obtained in the literature for other objects and conditions.

Author Response

(The authors gave the same response as above.)

Reviewer 3 Report

Comments and Suggestions for Authors

Dear authors,

I am extremely glad to have been chosen to review such an interesting paper as yours. The paper presents in a special and very detailed way the problem caused by climate change, which manifested itself as heavy rainfall event.  In addition to others, the sociological and economic consequences of such events, which have a very difficult effect on the population in cities, are particularly highlighted. For this reason, the paper presents models for solving this problem, which will represent an event of frequent repetition in the future.

All parts of the paper are written clearly and precisely, the paper contains all necessary elements such as introduction, material and method of work, presentation of results and conclusion. As a reviewer, I noticed that the main shortcoming of the work is in the discussion, which is insufficiently and inadequately written and processed. There are no examples from practice that would be compared with the proposed solutions, so I would ask you to correct your work in that direction.

Good luck!

Best regards!

Author Response

(The authors gave the same response as above.)

Round 2

Reviewer 1 Report

Comments and Suggestions for Authors

I have carefully rechecked the revised manuscript (sustainability-2986632), and found that this is the shortest and simplest response I have ever seen. Many major serious problems in the first version have not been resolved.

- . Although the title emphasizes "sustainability", in actual discussions, considerations of sustainability seem to focus mainly on technology and engineering, and lack comprehensive consideration of multiple dimensions such as economy, society and environment. A truly sustainable drainage system should be able to balance the needs of these aspects.

- . The study focused on the city of Urmia, a region that may have specific geographical, climatic and socioeconomic conditions. Therefore, the results may not be directly applicable to other cities or regions with different characteristics. In addition, there is a lack of comparison with cities with similar climatic conditions and geographical locations.

- . The authors suggest that vegetative swales are the most promising solution. However, this singular recommendation may overlook other potentially effective measures, such as green roofs and rain gardens, which may be more applicable under different conditions.

- . The study has mainly focused on technical solutions, but has not adequately considered socio-economic factors (such as cost-benefit analysis, public acceptance, policy environment, etc.).

- . The evaluation of these scenarios is not comprehensive. For example, there is no in-depth discussion and comparison of the long-term effects, economic costs, environmental impacts, and social acceptance of different scenarios.

- . The study is mainly based on model simulations and lacks field verification.

- . The literature review section is not even enough. There are a number of studies that have considered land use in flood mitigation. This aspect has been ignored in the manuscript. Please see below for example.

Assessing the scale effect of urban vertical patterns on urban waterlogging: An empirical study in Shenzhen, 2024, 107486.

From risk control to resilience: developments and trends of urban roads designed as surface flood passages to cope with extreme storms. Front. Environ. Sci. Eng. 18, 22

- . The study has mainly focused on the design and effectiveness of drainage systems in the short term, but there is a lack of in-depth discussion and evaluation of the impact of climate change on urban drainage systems and the adaptability of drainage systems in the long term. With the intensification of climate change, this aspect of evaluation is particularly important.

- . The authors should not just post the screenshots of the software in the manuscript.

Comments on the Quality of English Language

Moderate editing of English language required.

Author Response

Thank you for your valuable feedback on my manuscript. I have carefully addressed all your comments and suggestions in the revised version. The final manuscript has been delivered for your review. The modifications are highlighted gray in the manuscript.

I appreciate your time and effort in improving the quality of this work.

Reviewer 3 Report

Comments and Suggestions for Authors

Dear authors,

thanks for finishing the work and upgrading the discussion.

However, discussion involves comparing your results of this research with the results of other reference authors. I would like you to supplement the discussion in this sense.

Best regards

Author Response

(The authors gave the same response as above.)

Round 3

Reviewer 1 Report

Comments and Suggestions for Authors

Thank you for incorporating my comments. I am pleased to recommend publication of this paper.